# DOVTrack: Data-Efficient Open-Vocabulary Tracking

**Zekun Qian[1,3], Ruize Han[2]†, Zhixiang Wang[1], Junhui Hou[3], Wei Feng[1]**
[1]College of Intelligence and Computing, Tianjin University,
[2]Shenzhen University of Advanced Technology, [3]City University of Hong Kong
{clarkqian, zhixiang_wang, wfeng}@tju.edu.cn,

hanruize@suat-sz.edu.cn, jh.hou@cityu.edu.hk

## Abstract

Open-Vocabulary Multi-Object Tracking (OVMOT) aims to detect and track multi-category objects including both seen and unseen categories during training. Currently, a significant challenge in this domain is the lack of large-scale annotated video data for training. To address this challenge, this work aims to effectively train the OV tracker using only the existing limited and sparsely annotated video data. We propose a comprehensive training sample space expansion strategy that addresses the fundamental limitation of sparse annotations in OVMOT training. Specifically, for the association task, we develop a diffusion-based feature generation framework that synthesizes intermediate object features between sparsely annotated frames, effectively expanding the training sample space by approximately 3× and enabling robust association learning from temporally continuous features. For the detection task, we introduce a dynamic group contrastive learning approach that generates diverse sample groups through affinity, dispersion, and adversarial grouping strategies, tripling the effective training samples for classification while maintaining sample quality. Additionally, we propose an adaptive localization loss that expands positive sample coverage by lowering IoU thresholds while mitigating noise through confidence-based weighting. Extensive experiments demonstrate that our method achieves state-of-the-art performance on the OVMOT benchmark, surpassing existing methods by 3.8% in TETA metric, without requiring additional data or annotations. The code will be available at https://github.com/zekunqian/DOVTrack.

## 1 Introduction

Open-Vocabulary Multi-Object Tracking (OVMOT) aims to track objects of any given category within a scene, including unseen classes during training [1]. Unlike traditional Multi-Object Tracking (MOT) tasks, OVMOT is not limited to specific categories of objects, such as pedestrians and vehicles; instead, it can handle a broader range of object categories. This capability enhancement significantly improves the applicability of tracking problems but introduces greater challenges. One major challenge is building datasets that are both large-scale and diverse in object categories. However, most existing large MOT datasets [2–6] often suffer from limited categories, making it difficult to train effective OVMOT algorithms.

To obtain effective training data, most current approaches [1, 7, 8] primarily rely on large-scale image datasets like LVIS [9], which facilitates OVMOT training by augmenting single images into image pairs. While such image data can supplement target categories and increase annotation scale, they are fundamentally limited by their static nature and lack of temporal information. This limitation restricts the model's ability to learn essential dynamic features of videos, such as object deformations, viewpoint changes, and gradual occlusion. Therefore, while modern OVMOT research relies on these image datasets, training a robust OV tracker using continuous video data appears more promising.

---

†Corresponding author.

39th Conference on Neural Information Processing Systems (NeurIPS 2025).

Recently, the TAO dataset [10], with its 833 categories, is the only suitable video training set that meets the diversity requirements for current OVMOT research. Although TAO offers a rich variety of categories, its small size and sparse annotations pose significant challenges for model training. Specifically, although TAO consists of 500 videos, one annotated frame is provided every 30 frames, resulting in an average of only 37 annotated frames per video. Additionally, each frame has a limited number of annotated targets, with an average of only 2 annotated objects per frame. This spatiotemporal sparse annotation format and limited data scale make it difficult for the model to effectively capture temporal dynamic changes of the targets and the spatially dense objects in each frame, thus leading to challenges in OVMOT training. As validated by experiments in OVTrack [1] and SLAck [11], using such a small-scale dataset with sparse annotations cannot be used to train a usable OV tracker. This leads us to a key question: *Is it possible to train an effective and robust OV tracker directly using the TAO dataset with only limited and sparsely annotated data?* This problem is challenging for two key reasons. First, it is essential to effectively utilize the temporally sparse annotated data to obtain more continuous target features at intervals (un-annotated frames), thereby enhancing the continuity of the tracking training. Second, the training is constrained by the limited size of data, necessitating the learning of sufficient meaningful information from the small-scale data for efficient training.

For the temporal sparse annotation issue, the sole work, SLAck [11], has attempted to train an OVMOT using the TAO dataset. It utilizes annotations from previous frames and the targets detected at intervals to generate pseudo labels based on Intersection over Union (IoU) for training. This method aims to convert the originally sparse annotations in the videos into a denser format, enhancing the continuity of the video annotations and increasing the number of learnable samples. However, this approach has notable limitations: First, pseudo labels built on previous frames are only effective for adjacent frames. Those generated at longer intervals often exhibit significant deviations, especially for rapidly moving targets. Second, this method relies on frame-by-frame processing of un-annotated images and on the quality of the detector, resulting in time-consuming preprocessing and unstable performance. Third, this approach computes IoU only with a single annotated frame to predict annotations, without considering relationships with neighboring annotated frames. It focuses solely on unidirectional continuity and fails to align pseudo labels generated within intervals with annotations on both sides.

To address these challenges, we aim to develop an implicit approach to directly obtain the target features at un-annotated intervals. Inspired by the recent success of diffusion models [12] in pixel-level generation, we explore the potential of applying generative models to feature-level generation. Instead of relying solely on the final denoised result at the starting point, we consider using the intermediate results obtained during the denoising process as usable outputs. This leads to a novel diffusion-based approach for generating object features at the intervals from sparse annotations. As illustrated in Figure 1(a). For the first time, we

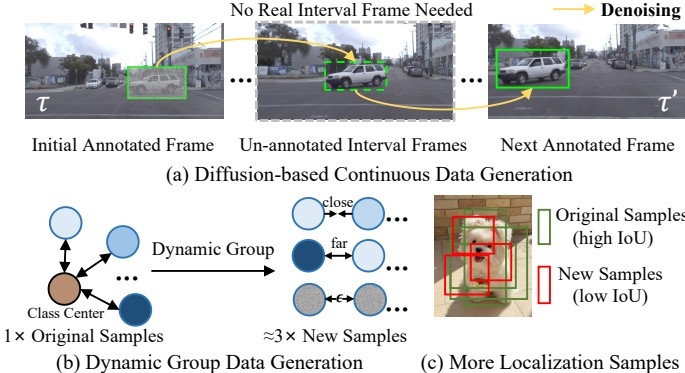

(a) Diffusion-based Continuous Data Generation

(b) Dynamic Group Data Generation

(c) More Localization Samples

Figure 1: Illustration of our strategies for effectively enriching trainable samples. (a) Generating unlabeled object features at intervals using a diffusion model. (b) Generating approximately three times more samples via a dynamic grouping strategy. (c) Harvesting additional low-confidence samples by lowering the IoU threshold between detections and ground truth.

model the denoising process of diffusion as the temporal evolution of target features, simulating the changes in target features between the two sparse annotated frames to acquire the object features in the intervals. Specifically, we construct the starting point of the diffusion model's denoising process by adding Gaussian noise $\epsilon$ to the object features at the initial annotated frame $\tau$. We then perform denoising at various time steps to learn the target features at the interval frames. The output of the denoising process is the next annotated frame $\tau'$. To ensure the reliability of the object features during interval frames, we develop 1) a reconstruction loss that utilizes interpolation information as supervision to ensure the feature reliability; 2) an endpoint loss to guarantee that the endpoint of the

denoising path accurately aligns with the object from $\tau'$; 3) a smoothness loss that balances feature smoothness between un-annotated frames and neighboring annotated frames to achieve smooth object feature transitions. By training the diffusion model, we simulate the object feature variations and obtain the object features at intervals between two annotated frames. These features will subsequently be used for training, addressing the challenges of effective training with sparse annotations.

Furthermore, most existing detection-based OVMOT methods directly use LVIS pre-trained Open-Vocabulary Detector (OVD) and do not consider OVD during the tracker training process. To investigate thoroughly, we first find that fine-tuning the pre-trained OVD on small-scale datasets, like TAO with long-tailed category distributions, leads to a significant decline in performance. The primary reason for this issue is that models pre-trained on large-scale datasets, like LVIS (164K samples), struggle to effectively adapt to the smaller and unevenly distributed samples in datasets like TAO. Since detection is very important for OVMOT, we explore how to effectively fine-tune the OVD using the TAO with small-scale data and annotations. For this purpose, we propose a couple of new strategies to address the issue of insufficient learnable samples in TAO, thereby improving object classification and localization performance. First, as shown in Figure 1(b), we design a dynamic group contrastive learning strategy that categorizes object features into affinity groups, dispersion groups, and adversarial groups, respectively, to enhance the object category diversity. With this strategy, the available samples have increased approximately threefold, effectively improving the contrastive learning for object classification under sparse and limited data conditions. It also effectively increases category diversity by adding the implicit new samples. Additionally, we introduce an adaptive localization loss strategy. Specifically, as shown in Figure 1(c), by deliberately lowering the sampler IoU threshold between detection boxes and ground truth (GT), we first incorporate more low-confidence positive samples. Then, for the localization task, we dynamically adjust loss weights to make full use of the low-confidence samples, effectively mining the information of valid samples. These two strategies significantly increase the quantity of learnable samples, resulting in higher accuracy and robustness in the model's localization and classification tasks. Our contributions can be summarized as follows:

1. Addressing the OVMOT association training problem under sparse annotations: We propose a novel diffusion-based model to simulate the features of targets at (un-annotated) interval moments, effectively enriching the feature space by enhancing the continuity of association features, thereby overcoming key challenges posed by temporal sparse annotations.

2. Achieving efficient fine-tuning of OV detectors under small-scale data: We propose a new dynamic group contrastive learning strategy to improve the classification by implicitly constructing new training samples. We also propose an adaptive localization loss that not only significantly expands the sample size but also alleviates the influence of low-confidence samples for localization learning.

3. Achieving state-of-the-art (SOTA) performance on the OV-TAO benchmark: Extensive results show that the proposed method achieves SOTA results, and verify the effectiveness and superiority of each component in our method.

## 2 Related Work

**Open-world/vocabulary object detection.** Open-world object detection diverges from traditional approaches by discovering salient objects without relying on a predefined label set [13–15]. Rather than assigning instances to known categories, it formulates recognition as clustering around learned class prototypes, enabling the discovery of novel objects [14]. Open vocabulary detection (OVD) takes this a step further by requiring explicit prediction of unseen class names [16], commonly achieved by integrating text embeddings into the detector's training pipeline [17, 18]. The emergence of vision–language models like CLIP [19], which align visual features with textual descriptions, has markedly enhanced open-vocabulary classification. Building on CLIP, recent studies [20–22] adapt these pre-trained models for both open-vocabulary and few-shot object detection. Furthermore, prompt-learning techniques that refine class-description embeddings [23–25] have been shown to boost detection accuracy in open-vocabulary scenarios. Unlike existing OVMOT approaches that simply deploy a pretrained OVD in downstream tasks to avoid performance loss, our method explicitly addresses the degradation in detection performance that occurs when fine-tuning OVD on small-scale datasets, substantially enhancing both detection and classification accuracy.

**Open-world/vocabulary object tracking.** Research on open-world tracking remains limited. Early methods either segment or follow every moving object in a video [26, 27] or employ class-agnostic detectors for generic object tracking [28–30]. The TAO-OW benchmark [31] is introduced to drive progress in this area, but it evaluates only class-agnostic metrics and overlooks class-specific performance. To address these gaps, OVTrack [1] brings open-vocabulary capabilities to tracking by integrating OVD into a framework to recognize a wide variety of scene objects. It also establishes a new baseline and benchmark built upon the TAO dataset. However, most existing OVMOT methods [1, 32, 8] rely on a frozen OVD and train the association head exclusively on large sets of static image pairs, thereby ignoring the temporal continuity present in video data. In contrast, our approach achieves effective OVD fine-tuning with only a few samples and introduces a diffusion-based target-feature association learning algorithm that attains SOTA performance using merely sparsely annotated frames. Moreover, when compared to other TAO-trained OVMOT algorithms [11], *i.e.*, SLAck, our method surpasses its tracking accuracy despite not using any interval frames. While SLAck augments appearance features with motion and classification cues, we still exceed its performance by relying solely on appearance information.

**Diffusion model.** The diffusion model is a recently popular class of deep-learning-based generative models that recover valid samples from a random distribution via an iterative denoising process. It has achieved remarkable success in image generation, *e.g.*, DALL·E 2 [33], Stable Diffusion [34] and VQ-Diffusion [35], *etc.* Following the trajectory of the success of image generation, video generation methods [36–40] have also made significant progress, using text prompts to guide diffusion models in producing coherent frame sequences. However, existing pixel-level video diffusion models demand large size of data to generate intermediate frames, incurring substantial computational costs and training time. They also cannot generate accurate annotations for those synthesized frames, leaving the problem of learning from sparsely labeled datasets unaddressed. To address this problem, we propose a feature-level diffusion framework that models object state transformation between sparsely annotated frames. Our method uses only a few sparse annotations to directly simulate each target's feature evolution between labeled frames, efficiently generating intermediate object representations and thus enabling high-performance OV tracker training.

## 3 Proposed Method

This work primarily focuses on addressing the challenges of training the OVMOT model with limited sparsely annotated data, rather than on the design of the model structure itself. Therefore, we utilize the baseline architecture from OVTrack [1], which is built upon the ResNet-50 backbone and includes the localization, classification and association heads as below:

**Localization**: It employs a class-agnostic object proposal approach from Faster R-CNN [41] to localize objects for both base and novel categories, generating bounding boxes and confidence scores.

**Classification**: The classification head enhances the framework's open-vocabulary capabilities through feature distillation with CLIP [19]. It obtains classification features $F_{cls}$ of objects and aligns them with the text features $F_{text}$ generated by CLIP's text encoder. By calculating the cosine similarity between these features and predefined novel class names, the framework determines if the similarity exceeds a threshold, thereby identifying the corresponding novel classes.

**Association**: The association head links detected objects across frames by learning object appearance features $F_{asso}$ for similarity measurements. If the appearance similarity between objects exceeds a certain threshold, they are considered the same target and assigned to the same tracking trajectory.

### 3.1 Diffusion-based Data Generation for Association Training

Inspired by the success of diffusion models in various generative tasks, we aim to leverage this framework for feature association in OVMOT. Classical diffusion methods typically focus on obtaining a final denoised feature representation. However, our approach takes a novel perspective by utilizing intermediate results throughout the diffusion process instead of solely relying on the final output.

As shown in Figure 2, given the object features from one annotated frame $\tau$ as the key feature $F_{key} \in \mathbb{R}^{B \times d}$. We denote the features of the same object from the next annotated frame $\tau'$ as the reference feature $F_{ref} \in \mathbb{R}^{B \times d}$, where $B$ is the batch size and $d$ is the dimension of the features. We model the denoising process as learning the mapping from $F_{key}$ to $F_{ref}$. The aim is to utilize the

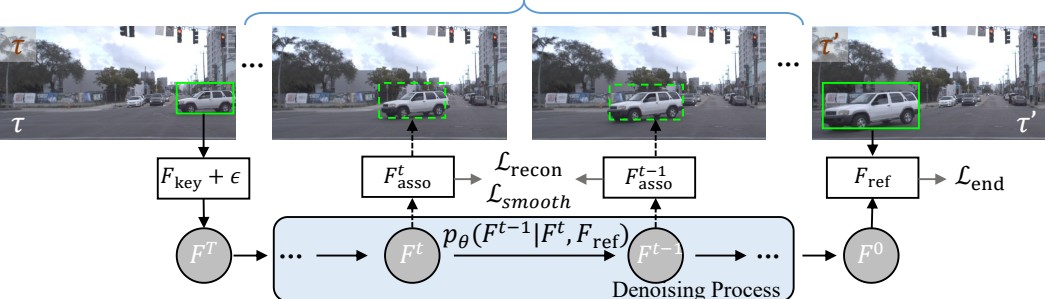

Figure 2: Illustration of the denoising process in our diffusion model, which simulates the transformation of object features from $F_{\text{key}}$ to $F_{\text{ref}}$ to obtain the un-annotated object features at interval frames.

diffusion model to simulate the intermediate states of the object and extract associated features at different timesteps.

**Forward Process.** The forward noise perturbing process at time $t$ is defined as

$$q_\theta(F^t|F^{t-1}) = \mathcal{N}(F^t; \sqrt{1-\beta_t}F^{t-1}, \beta_t\mathbf{I}), \tag{1}$$

which gradually adds Gaussian noise to the data according to a variance schedule $\beta_1, \ldots, \beta_T$. The construction of features at time $t$ can be expressed as $F^t = \sqrt{\alpha_t}F_{\text{key}} + \sqrt{1-\alpha_t}\epsilon$, where $\alpha_t = \prod_{s=1}^{t}(1-\beta_s)$ and $\epsilon \sim \mathcal{N}(0, \mathbf{I})$. Here, $\alpha_t$ represents the cumulative product of the variance schedule up to time $t$.

**Reversed Process.** The reverse process $p_\theta(F^{t-1}|F^t, F_{\text{ref}})$ then simulates the transition from $F_{\text{key}}$ to $F_{\text{ref}}$, systematically obtaining $F_{\text{ref}}$ from $F^t$ through the denoising process, as

$$F^{t-1} = \frac{1}{\sqrt{\alpha_t}}\left(F^t - \frac{1-\alpha_t}{\sqrt{1-\prod_{i=1}^{t}\alpha_i}}f_\theta(F^t, F_{\text{ref}}, t)\right), \tag{2}$$

where $f_\theta$ is a neural network that estimates the noise present in $F^t$, incorporating $F_{\text{ref}}$ as a crucial input for refining its predictions.

**Loss Functions.** To ensure effective training of the diffusion model, we define the following loss functions that encapsulate the mapping process from $F_{\text{key}}$ to $F_{\text{ref}}$. In the following, we denote the intermediate features $F^t$ as $F_{\text{asso}}^t$, corresponding to the association features in OVMOT at timestep $t$.

**1) Reconstruction loss.** To ensure that the features at frame $t$ accurately represent the changes between $F_{\text{key}}$ and $F_{\text{ref}}$, we construct a reconstruction loss through interpolation. We use time $t$ to balance the interpolation ratio, resulting in the following loss

$$\mathcal{L}_{\text{recon}} = \frac{1}{N}\sum_{t\in\{t_1, t_2, \ldots, t_N\}}\|F_{\text{asso}}^t - [t'F_{\text{key}} + (1-t')F_{\text{ref}}]\|_2^2, \tag{3}$$

where $t_i$ are the timesteps, $N$ is the number of sampled timesteps, and $t' \in [0, 1]$ is the normalized timesteps. This loss encourages the model to reconstruct $F_{\text{asso}}^t$ as a linear interpolation between $F_{\text{key}}$ and $F_{\text{ref}}$ based on the current normalized timestep $t'$, reinforcing the model to approximate the state between the key and reference features.

**2) Endpoint loss.** In addition to ensuring the accuracy of the reconstructed features at intermediate timesteps, it is also essential that the final denoised features align with $F_{\text{ref}}$ to establish a complete transformation chain from $F_{\text{key}}$ to $F_{\text{ref}}$, resulting in the following loss

$$\mathcal{L}_{\text{end}} = \|F_{\text{asso}}^0 - F_{\text{ref}}\|_2^2. \tag{4}$$

This loss ensures that the final associated features $F_{\text{asso}}^0$ (resulting from the complete denoising process) exactly match the reference features $F_{\text{ref}}$.

**3) Smoothness loss.** Although the two losses above ensure the accuracy of the results at each timestep, they do not account for the smoothness of the feature changes at each timestep, while the

objects always change smoothly in a continuous video. To address this, we consider the consistency relationship between features during the transformation process and construct the following loss

$$\mathcal{L}_{\text{smooth}} = \left\| \frac{F_{\text{asso}}^t - F_{\text{key}}}{\|F_{\text{asso}}^t - F_{\text{key}}\|} - \frac{F_{\text{ref}} - F_{\text{asso}}^t}{\|F_{\text{ref}} - F_{\text{asso}}^t\|} \right\|_2^2, t \in \{t_1, t_2, \ldots, t_N\}. \tag{5}$$

This loss encourages smooth transitions between the associated features across timesteps, preventing abrupt changes in feature representation, which can occur due to the inherent noise in observed data.

Overall, these losses assist the diffusion model in constructing accurate and smooth feature representations at intermediate timesteps, facilitating the overall mapping from $F_{\text{key}}$ to $F_{\text{ref}}$. The intermediate features $F_{\text{asso}}^t$ obtained during the denoising process of the diffusion model will be utilized as the object association features at un-annotated interval frames for association training, addressing the issue of sparse annotations.

**Enhanced Association Training with Generated Features.** Our diffusion model fundamentally transforms sparse association training by generating intermediate features $F_{\text{asso}}^t$ at multiple timesteps between annotated frames. This enhancement operates through two complementary mechanisms:

**1) Quantitative Sample Space Expansion.** For each object trajectory originally containing only two annotated features ($F_{\text{key}}$ and $F_{\text{ref}}$ separated by 30 frames), our method generates $N$ intermediate features where $N$ corresponds to sampling steps (typically 3). This directly expands the positive sample set $Q^+(\mathbf{q})$ for each object identity from 2 to 5 features, while simultaneously enriching the negative sample set $Q^-$ with generated features from different object trajectories. Consequently, the total training samples increase by approximately 3×, providing substantially more positive and negative pairs for robust contrastive learning.

**2) Qualitative Feature Continuity Enhancement.** Beyond quantity expansion, our generated intermediate features $F_{\text{asso}}^t$ exhibit superior temporal continuity compared to sparse annotations. The diffusion-based generation ensures smooth feature transitions through: (1) reconstruction loss enforcing accurate interpolations between keyframes; (2) smoothness loss guaranteeing consistent feature evolution; (3) endpoint loss ensuring precise alignment with reference features. This results in temporally coherent feature representations that better capture object state transitions, providing higher-quality training samples for association learning.

Building upon these enhancements, we employ the same association loss framework as OVTrack [1] but with significantly enhanced effectiveness due to our enriched sample space. The complete tracking loss $\mathcal{L}_{\text{track}}$ is formulated as:

$$\mathcal{L}_{\text{track}} = -\sum_{\mathbf{q} \in Q} \frac{1}{|Q^+(\mathbf{q})|} \sum_{\mathbf{q}^+ \in Q^+(\mathbf{q})} \log \left( \frac{\exp(\mathbf{q} \cdot \mathbf{q}^+/\tau)}{\text{PosD}(\mathbf{q}) + \sum_{\mathbf{q}^- \in Q^-(\mathbf{q})} \exp(\mathbf{q} \cdot \mathbf{q}^-/\tau)} \right), \tag{6}$$

where $\text{PosD}(\mathbf{q}) = \frac{1}{|Q^+(\mathbf{q})|} \sum_{\mathbf{q}^+ \in Q^+(\mathbf{q})} \exp(\mathbf{q} \cdot \mathbf{q}^+/\tau)$ and $\tau$ is the temperature parameter. The enriched $Q^+(\mathbf{q})$ contains both original keyframe features and our generated intermediate features, while $Q^-(\mathbf{q})$ includes negative samples from different trajectories.

## 3.2 Training Sample Extending for Detection

Besides association, we next consider how to train (fine-tune) the detection model using the limited data in TAO, which includes both the classification and localization heads. Given that TAO is a long-tailed dataset with limited annotations, we aim to address two main challenges. First, for classification, we need to overcome the limitations of sample quantity and diversity to effectively learn class representations from a small number of samples. This way, we propose a dynamic group contrastive learning approach, which increases the number of samples used in contrastive learning by approximately 3 times, thereby significantly enhancing classification training. Second, for localization, we seek to increase the number of localization samples (bounding boxes) by lowering the IoU threshold for positive samples, and we apply adaptive confidence weights to reduce the noise from low-confidence samples. We will elaborate on the methods in detail below.

**Dynamic Group Contrastive Learning.** We present the proposed dynamic group contrastive learning from 1) how to construct dynamic groups and why dynamic groups effectively enhance sample quality, and 2) how to utilize the generated dynamic groups for contrastive learning.

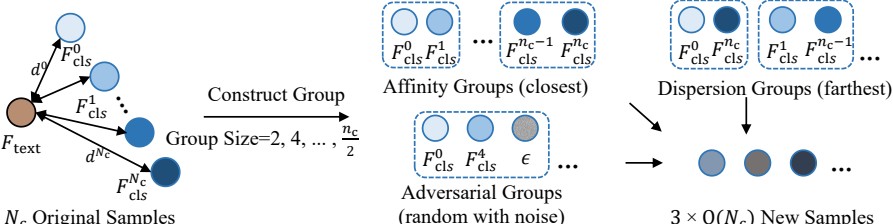

Figure 3: Illustration of the construction of our dynamic groups, which generates diverse and robust group samples, effectively tripling the training samples. For clarity, the diagram depicts only the construction process for a group size of two; other group sizes follow the same principles.

**1) Dynamic group construction.** As shown in Figure 3, given the classification features (obtained from the classification head as discussed above) of all $N_c$ samples in each class, for each sample $i$, we first calculate the Euclidean distance $d^i$ to its corresponding text center features $F_{\text{text}}$ of this class. Then, we sort the samples in ascending order of distance, resulting in $F_{\text{cls}}^0, F_{\text{cls}}^1, \ldots, F_{\text{cls}}^{N_c-1}, F_{\text{cls}}^{N_c}$. This sorting process ranks the samples according to their similarity to the CLIP text embedding of this class, which reflects the difficulty of each sample. We then group these samples in each class by dividing all samples into $g$ groups, where $g = 2, 4, 8, 16, \ldots$ up to $2^{\lfloor \log_2 N_c - 1 \rfloor}$, with $N_c$ as the total number of samples. The number of groups follows a geometric progression, where each number is double the previous one. The total count of groups formed can be derived from the geometric sum formula, *i.e.*, $2^{\lfloor \log_2 N_c - 1 \rfloor + 1} - 2$, with an order of magnitudes of $O(N_c)$. After determining the groups, each group aggregates the features of its respective samples, which is achieved by the averaging process. To enhance the diversity and robustness of the newly constructed groups, we have designed three distinct types of grouping manners:

❶ Affinity group: The samples are sorted according to their distances to the corresponding text center. After that, each group consists of several samples that are closest to each other. We achieve this by minimizing the sum of variances of all $g$ groups. The aggregated features from each similarity group serves to reinforce the shared features among similar samples, thereby improving the model's ability to distinguish fine-grained differences.

❷ Dispersion group: Similar to the above operation, instead of clustering the closest samples, this manner selects the mutually dispersed samples within each class as much as possible. This is achieved by maximizing the sum of variances of all $g$ groups. By doing so, each dispersion group ensures a mix of representative samples that capture varying aspects of the feature space. This manner enhances the model's exposure to a broader range of features, ultimately improving its generalization capabilities. The detailed implementation and theoretical proof of grouping process are provided in Appendix A.

❸ Adversarial group: In constructing the adversarial group, samples are randomly selected from the pool without considering their similarity-based ordering. Following the selection, Gaussian noise is added to the features of the chosen samples to create adversarial conditions. This randomness and the addition of noise help to simulate challenging scenarios for the model, forcing it to learn more robust feature representations that can withstand variations and perturbations during testing.

**2) Contrastive learning with augmented samples.** With the enhanced dataset containing $N_c + 3$ $O(N_c)$ samples, which includes the $N_c$ original samples and the 3 $O(N_c)$ newly generated samples from 3 types of groups, we can perform contrastive learning to improve model robustness and feature representation. The overall training objective for this augmented dataset is to minimize the InfoNCE loss [42] formulated as follows:

$$\mathcal{L}_{\text{InfoNCE}} = -\frac{1}{N} \sum_{i=1}^{N} \log \frac{\exp(\cos(F_{\text{cls}}^i, F_{\text{cls}}^{i+})/\rho)}{\sum_{k=1}^{K} \exp(\cos(F_{\text{cls}}^i, F_{\text{cls}}^k)/\rho) + \exp(\cos(F_{\text{cls}}^i, F_{\text{cls}}^{i+})/\rho)}, \quad (7)$$

where $N$ is the total number of samples, *i.e.*, $N_c + 3$ $O(N_c)$, $F_{\text{cls}}^i$ is the feature representation of the $i$-th sample, $F_{\text{cls}}^{i+}$ is the positive sample paired with $F_{\text{cls}}^i$, $K$ is the total number of negative samples in the batch, $\cos(\cdot, \cdot)$ denotes the cosine similarity function, $\rho$ is a temperature parameter that controls the distribution sharpness. The InfoNCE loss encourages the model to bring the same class samples closer together in the feature space while pushing different class samples further apart. By leveraging the increased sample size from the augmented groups, the effectiveness of this loss

function is enhanced, improving the model's ability to discern differences between classes. For scalability analysis of this approach with large category sets, please refer to Appendix B.4.

**Adaptive Localization Loss.** In object localization subtasks of OVD, it is a common operation to compute losses only for the positive samples that exceed a certain IoU threshold with the ground truth (GT). This is acceptable for training on large-scale datasets. For limited data, to enlarge the sample size, we propose to increase the number of positive training samples by lowering the IoU threshold, but this also introduces additional disturbance into the training process. To address this issue, we implement an adaptive localization loss that considers the confidence of each sample.

We define the bounding box for the $i$-th sample as $B_i = (x_i, y_i, w_i, h_i)$, with $x_i$ and $y_i$ as the coordinates of the top-left corner and $w_i$ and $h_i$ as the width and height, respectively. The corresponding ground truth bounding box is denoted as $B_i^*$. To manage the disturbance created by the low-IoU bounding boxes, we employ a smoothing weight strategy defined for the $i$-th object as $w_i = \frac{1}{\log(\epsilon_i + 1) + 1} \in (0, 1]$, where $\epsilon_i = ||B_i - B_i^*||$ denotes the absolute error between the predicted and ground truth bounding box. This smoothing weight strategy assigns lower weights to the boxes with larger errors. The intuition behind this approach is that predictions that significantly deviate from the ground truth are often less reliable. By reducing their impact on the overall loss, we extend the sample variety while emphasizing more accurate predictions, which positively contributes to the training process. The overall adaptive localization loss can then be formulated as

$$\mathcal{L}_{\text{Adaptive}} = \frac{1}{N_{\text{p}}} \sum_{i=1}^{N_{\text{p}}} w_i \cdot L_{\text{SmoothL1}}(B_i, B_i^*), \tag{8}$$

where $N_{\text{p}}$ is the total number of positive samples and we use the Smooth $L_1$ loss in Faster R-CNN [41]

$$L_{\text{SmoothL1}}(B_i, B_i^*) = \begin{cases} 0.5||B_i - B_i^*||^2 & \text{if } ||B_i - B_i^*|| < 1 \\ ||B_i - B_i^*|| - 0.5 & \text{otherwise.} \end{cases} \tag{9}$$

This formulation of the adaptive localization loss effectively balances the inclusion of more training samples and the necessity of disturbance management. By applying adaptive weights based on the confidence of each prediction, we enhance the model's robustness and accuracy during training, ultimately improving its performance on the object localization subtask.

### 3.3 Implementation Details

We adopt the OVTrack [1] architecture and replicate its original pre-training protocol. In the proposed TAO training stage, we jointly optimize the association, localization and classification branches on the base set of the TAO training set. Training is performed for 10 epochs on only 2 RTX 3090 GPUs. In the association training, we employ a D$^2$MP-based diffusion model to denoise samples drawn from a standard normal distribution. $f_\theta$ is implemented as a three-layer fully-connected network, with each layer followed by a ReLU activation and layer normalization. During the first five epochs, the diffusion model is trained on features produced by the association head (without generating new samples), and in the subsequent 5 epochs, we freeze the diffusion model and use it to generate augmented data for further association training. The association loss, consisting of a contrastive loss and an auxiliary loss, is identical to that used in OVTrack. In the classification training, we apply an InfoNCE loss with $\rho = 0.1$ in Eq. 7 and also employ the standard cross-entropy loss used in OVTrack. In the localization training, the IoU threshold is lowered to 0.3. In the inference stage, we retain all OVTrack settings except that we change the maximum detections per frame to 80, the matching threshold to 0.38 and the memory length to 30.

## 4 Experimental Results

### 4.1 Datasets and Metrics

Following other OVMOT methods [1, 11, 7, 8], we perform our evaluation with standard OV settings on the TAO dataset, which categorizes rare classes as novel and the others as base classes, similar to LVIS [9]. Comparative experiments are carried out on both the validation and test sets of TAO. For performance assessment, we implement the standard OVMOT metric tracking-everything accuracy (TETA) [43], which encompasses evaluation of localization accuracy (LocA), classification accuracy

Table 1: Comparison of tracking performance on validation and test sets of the open-vocabulary TAO benchmark [1]. All methods use ResNet-50 as the backbone. † represents using the same detector.

| Method | Novel | | | | Base | | | |
|---|---|---|---|---|---|---|---|---|
| | TETA | LocA | AssocA | ClsA | TETA | LocA | AssocA | ClsA |
| **Validation set** | | | | | | | | |
| QDTrack [44] | 22.5 | 42.7 | 24.4 | 0.4 | 27.1 | 45.6 | 24.7 | 11.0 |
| TETer [43] | 25.7 | 45.9 | 31.1 | 0.2 | 30.3 | 47.4 | 31.6 | 12.1 |
| DeepSORT (ViLD) [45] | 21.1 | 46.4 | 14.7 | 2.3 | 26.9 | 47.1 | 15.8 | 17.7 |
| Tracktor++ (ViLD) [46] | 22.7 | 46.7 | 19.3 | 2.2 | 28.3 | 47.4 | 20.5 | 17.0 |
| ByteTrack† [47] | 22.0 | 48.2 | 16.6 | 1.0 | 28.2 | 50.4 | 18.1 | 16.0 |
| OC-SORT† [48] | 23.7 | 49.6 | 20.4 | 1.1 | 28.9 | 51.4 | 19.8 | 15.4 |
| OVTrack† [1] | 27.8 | 48.8 | 33.6 | 1.5 | 35.5 | 49.3 | 36.9 | 20.2 |
| MASA (R50)† [32] | 30.0 | 54.2 | 34.6 | 1.0 | 36.9 | 55.1 | 36.4 | 19.3 |
| OVTR [7] | 31.4 | 54.4 | 34.5 | 5.4 | 36.6 | 52.2 | 37.6 | 20.1 |
| OVSORT † [8] | 30.8 | 53.0 | 37.6 | 1.9 | 38.2 | 55.3 | 39.9 | 19.4 |
| SLAck† [11] | 31.1 | 54.3 | 37.8 | 1.3 | 37.2 | 55.0 | 37.6 | 19.1 |
| Ours† | **35.2** | **59.2** | **40.3** | **6.2** | **40.4** | **58.6** | **42.1** | **20.5** |
| **Test set** | | | | | | | | |
| QDTrack [44] | 20.2 | 39.7 | 20.9 | 0.2 | 25.8 | 43.2 | 23.5 | 10.6 |
| TETer [43] | 21.7 | 39.1 | 25.9 | 0.0 | 29.2 | 44.0 | 30.4 | 10.7 |
| DeepSORT (ViLD) [45] | 17.2 | 38.4 | 11.6 | 1.7 | 24.5 | 43.3 | 14.6 | 15.2 |
| Tracktor++ (ViLD) [46] | 18.0 | 39.0 | 13.4 | 1.7 | 26.0 | 44.1 | 19.0 | 14.8 |
| OVTrack† [1] | 24.1 | 41.8 | 28.7 | 1.8 | 32.6 | 45.6 | 35.4 | 16.9 |
| OVTR [7] | 27.1 | 47.1 | 32.1 | 2.1 | 34.5 | 51.1 | 37.5 | 14.9 |
| OVSORT † [8] | 28.1 | 48.0 | 33.4 | 2.7 | 35.1 | 51.6 | 38.3 | 15.4 |
| SLAck† [11] | 27.1 | 49.1 | 30.0 | 2.0 | 34.7 | 52.5 | 35.6 | 16.1 |
| Ours† | **30.1** | **53.0** | **33.5** | **3.9** | **38.7** | **56.6** | **41.8** | **17.7** |

(ClsA), and association accuracy (AssocA). For clear evaluation, we evaluate the performance of base and novel classes separately.

## 4.2 Comparison with State-of-the-Arts

We compare our method with recent tracking methods on both the validation and test sets of TAO. For a fair comparison, all methods utilize ResNet-50 as the backbone. We include closed-set baselines trained on all categories, established off-the-shelf trackers such as ByteTrack [47], OC-SORT [48], and MASA [32], as well as specialized OVMOT methods like OVTrack [1], the state-of-the-art methods, SLAck [11], OVSORT [8], and transformer-based OVTR [7].

As shown in Table 1, our method demonstrates a significant performance improvement over all competing methods in terms of the TETA metric on both the validation and test sets. Notably, we achieve a TETA score of **35.2%** on the validation set and **38.7%** on the test set, *surpassing the second-best method by 3.8% and 3.6%*, respectively. In terms of individual metrics, our method achieves particularly strong results in both novel and base AssocA. Specifically, we report a novel AssocA of **40.3%** and a base AssocA of **42.1%**, marking *significant increases of 4.9% and 4.5% over SLAck*, which is also trained on the TAO dataset. Note that we only use sparse annotation data, without the interval frames used in SLAck. These improvements highlight the efficacy of our approach for improving tracking performance. Additionally, we observe substantial improvements in both the ClsA and LocA metrics. These gains are particularly notable under limited data, where effective fine-tuning plays a crucial role in maximizing performance, which indicates that our proposed strategies for sample augmentation for OVD are highly effective. Importantly, although our approach uses the same architecture as OVTrack and is trained on sparse data, our results significantly outperform those of OVTrack. This demonstrates the high efficiency of our training strategy with limited, sparse data.

## 4.3 Ablation Study

**Effectiveness of diffusion-based data generation.** As shown in the Association module section of Table 2, the results indicate that the three proposed loss functions significantly enhance the association performance. Furthermore, consistent with the findings of SLAck [11], we observe that training solely on the original sparse TAO dataset, without supplementing intermediate frame object features, leads to poor association results. Additionally, we compared our approach with a direct linear interpolation data generation method. Although this method showed some performance improvement, the overall results remained inadequate. These ablation experiments demonstrate that the intermediate target features generated by our diffusion-based method substantially improve the association in OVMOT.

**Effectiveness of dynamic group contrastive learning.** As shown in the Classification module of Table 2, first, by removing some additional groups in the top three rows, we observe a decline in

Table 2: Ablation study results on the validation set. We compare the results of different ablation methods. Each module corresponds to a different aspects of association, classification and localization.

| Module | Ablation Method | Novel | | | | Base | | | |
|---|---|---|---|---|---|---|---|---|---|
| | | TETA | LocA | AssocA | ClsA | TETA | LocA | AssocA | ClsA |
| Association | w/o reconstruction loss | 34.3 | **59.4** | 38.9 | 4.7 | 39.3 | 58.6 | 39.7 | 19.7 |
| | w/o smoothness loss | 34.7 | 59.1 | 39.3 | 5.8 | 39.9 | 58.5 | 41.2 | 20.1 |
| | w/o endpoint loss | 34.8 | 59.2 | 39.6 | 5.6 | 39.8 | 58.6 | 40.7 | 20.1 |
| | w/o diffusion-based data generation | 31.1 | 58.1 | 31.7 | 3.6 | 36.7 | 58.3 | 33.5 | 18.2 |
| | linear interpolation data generation | 31.3 | 57.1 | 32.7 | 4.2 | 37.2 | 57.9 | 34.3 | 19.5 |
| Classification | w/o affinity group | 34.3 | 58.9 | 40.0 | 4.1 | 39.9 | **58.8** | 41.5 | 19.4 |
| | w/o dispersion group | 34.7 | 59.3 | 39.8 | 4.9 | 39.8 | 58.5 | 41.8 | 19.2 |
| | w/o adversarial group | 34.2 | 58.8 | 39.9 | 4.0 | 39.6 | 58.3 | 41.6 | 19.0 |
| | w/o extra group data on contrastive learning | 33.2 | 58.7 | 39.6 | 1.2 | 39.0 | 58.1 | 41.2 | 17.6 |
| | w/o dynamic group contrastive learning | 33.5 | 58.8 | 39.9 | 1.7 | 39.4 | 58.5 | 41.5 | 18.2 |
| Localization | w/o adaptive weight | 31.8 | 52.7 | 38.1 | 4.6 | 36.1 | 51.3 | 38.3 | 18.7 |
| | w/o lower sampler IOU | 32.3 | 54.7 | 37.6 | 4.7 | 38.1 | 55.8 | 39.9 | 18.5 |
| | Ours | **35.2** | 59.2 | **40.3** | **6.2** | **40.4** | 58.6 | **42.1** | **20.5** |

the final classification results. In the fourth row, we can see a significant impact when we perform contrastive learning using only the limited original samples, without providing the additional training samples derived from our groups. The results in this case are even worse than those in the fifth row, where we eliminate contrastive learning altogether and rely solely on cross-entropy loss. Additionally, the experiments in the fifth row further prove the effectiveness of our contrastive learning approach.

**Effectiveness of adaptive localization loss.** In the Localization section of Table 2, we can see that the proposed adaptive weights effectively reduce the impact of noise during localization training. Also, utilizing a lower IoU threshold allows us to obtain a greater number of available training samples.

**Impact of different sampling steps.** In Table 3, we examine the impact of different sampling steps on the results. Notably, even with a sampling step of just 1, there is a significant improvement compared to the approach without data generation. Although the results fluctuate with an increase in sampling steps, the differences are relatively small. This indicates that our proposed diffusion-based data generation method is quite stable when training with sparse data.

Table 3: Ablation study results of different sampling steps on the validation set.

| Sampling Steps | Novel | | | | Base | | | |
|---|---|---|---|---|---|---|---|---|
| | TETA | LocA | AssocA | ClsA | TETA | LocA | AssocA | ClsA |
| 1 | 33.2 | 58.0 | 37.4 | 4.3 | 38.7 | 57.8 | 38.4 | 19.9 |
| 2 | 34.8 | 59.1 | 40.4 | 4.9 | 40.2 | 58.3 | **42.3** | 20.1 |
| 3 | **35.2** | **59.2** | 40.3 | **6.2** | **40.4** | **58.6** | 42.1 | **20.5** |
| 4 | 35.1 | 58.7 | **41.5** | 5.2 | 40.1 | 58.4 | 42.2 | 19.8 |
| 5 | 35.1 | 59.8 | 40.5 | 5.1 | 40.2 | 58.6 | 42.0 | 20.0 |

Additional analysis on detection fine-tuning necessity, challenging scenarios, scalability, and association-centric metrics including IDF1 and MOTA are provided in Appendix B. More visualization results are provided in Appendix C.

# 5   Conclusion

In this work, we have proposed a data-efficient OVMOT method. Based on the existing OV video dataset with sparse and limited annotations, we develop a series of methods to explore higher data utilization. We consider three sub-tasks in OVMOT: for the association task, we develop a diffusion-based method for temporal object feature construction to generate intermediate features between sparsely annotated frames; for classification, we design a dynamic group construction method to increase the data diversity of each object class; for localization, we excavate more candidate boxes and adaptively use them. Experimental results verify the effectiveness of our method. Through this work, we hope to promote the efficient training of OVMOT and provide some insights for effective learning of other detection and tracking problems on limited data.

# Acknowledgment

This work was supported in part by the National Natural Science Foundation of China (NSFC) under Grant 62402490, 62422118, 62476196, 62572349; by the Emerging Frontiers Cultivation Program of Tianjin University Interdisciplinary Center; by the Hong Kong Research Grants Council under Grants 11219324, 11202320, and 11219422.

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

# A  Variance Implications of the Grouping Strategy

## A.1  Minimizing Variance Sum through Affinity Group Construction

**Problem Statement.** Given a set of numbers $x_1, x_2, \ldots, x_n$, we want to construct $k$ groups, denoted as $G_1, G_2, \ldots, G_k$, such that the sum of each group is $T_k$. The mean of each group is defined as $\mu_k = \frac{T_k}{m_k}$, where $m_k$ is the number of elements in group $k$. We aim to minimize the overall variance $S^2$ of the sample set can be expressed as:

$$S^2 = \frac{1}{k} \sum_{i=1}^{k} \left[ \frac{1}{m_k - 1} \sum_{j=1}^{m_k} (x_j - \mu_i)^2 \right].$$

**Strategy for Minimizing $S^2$.** To minimize $S^2$, we adopt a sequential grouping strategy as follows: First, sort the data values $x_1, x_2, \ldots, x_n$ in ascending order by the similarity from the $F_{\text{text}}$. Then divide the sorted data into $k$ contiguous groups. The first group contains the smallest values, while the last group contains the largest values.

**Proof.** Next, we will provide a detailed proof of why this affinity group construction strategy can effectively achieve the minimization of $S^2$ as follows:

- **Definitions and Setup**: Let the mean of each group be $\mu_k = \frac{T_k}{m_k}$, where $m_k$ is the number of elements in each group.

- **Exchange Argument**: Suppose there are two groups $i$ and $j$, with $\mu_i$ and $\mu_j$ as their respective means. Let there exist an element $x < y$ such that $x \in G_j$ and $y \in G_i$. Based on the grouping method, we can conclude that $T_i < T_j$, meaning that the sum of group $i$ is less than the sum of group $j$. We consider swapping these two elements:

$$T_i' = T_i - x + y,$$
$$T_j' = T_j - y + x.$$

The new group means after the swap will be:

$$\mu_i' = \frac{T_i'}{m_i} = \frac{T_i - x + y}{m_i},$$
$$\mu_j' = \frac{T_j'}{m_j} = \frac{T_j - y + x}{m_j}.$$

Next, we need to calculate the updated variances for groups $i$ and $j$:

$$S_{i'}^2 = \frac{1}{m_i - 1} \sum_{j=1}^{m_i} (x_j - \mu_i')^2,$$

$$S_{j'}^2 = \frac{1}{m_j - 1} \sum_{j=1}^{m_j} (y_j - \mu_j')^2.$$

- **Calculating the Change in Variance**: The change in the overall variance $\Delta S^2$ due to the swap is given by:

$$\Delta S^2 = S_{i'}^2 + S_{j'}^2 - (S_i^2 + S_j^2).$$

Calculating $\Delta S^2$ directly can be complex; however, by swapping the larger value $y$ from group $G_j$ with the smaller value $x$ from group $G_i$, we significantly alter the internal distribution of the elements within each group. The introduction of the larger value $y$ in group $G_i$ increases its variance because it increases the deviation from the new mean. Similarly, swapping the smaller value $x$ into group $G_j$ will also increase the variance of that group. Therefore, as a result of these changes, it can be inferred that:

$$\Delta S^2 > 0.$$

This indicates that the overall variance increases after any swap, which suggests that the process of selective grouping can minimize the spread of variance across the groups.

- **Conclusion:** Therefore, using a sequential grouping method (i.e., partitioning the data into contiguous segments) will minimize the overall variance $S^2$.

## A.2 Maximizing Variance Sum through Dispersion Group Construction

**Problem Statement.** Given a set of numbers $x_1, x_2, \ldots, x_n$, we want to construct $k$ groups, denoted as $G_1, G_2, \ldots, G_k$, such that the sum of each group is $T_k$. The mean of each group is defined as $\mu_k = \frac{T_k}{m_k}$, where $m_k$ is the number of elements in group $k$. The overall variance $S^2$ of the sample set can be expressed as:

$$S^2 = \frac{1}{k} \sum_{i=1}^{k} \left[ \frac{1}{m_k - 1} \sum_{j=1}^{m_k} (x_j - \mu_i)^2 \right].$$

Expanding this yields:

$$S^2 = \frac{1}{k} \sum_{i=1}^{k} \left[ \frac{1}{m_k - 1} \left( \sum_{j=1}^{m_k} x_j^2 - m_k \mu_i^2 \right) \right].$$

This can be rearranged to show a fixed term:

$$S^2 = \frac{1}{k} \sum_{i=1}^{k} \left[ \frac{1}{m_k - 1} \sum_{j=1}^{m_k} x_j^2 - \frac{m_k}{m_k - 1} \mu_i^2 \right].$$

The term $\sum_{j=1}^{n} x_j^2$ is a fixed quantity determined by the sample set. Therefore, to maximize $S^2$, we need to minimize the term: $\sum_{i=1}^{k} \mu_i^2$.

**Strategy for Minimizing $\sum_{i=1}^{k} \mu_i^2$.** To minimize $\sum_{i=1}^{k} \mu_i^2$, we adopt a two-end grouping strategy as follows: First, sort the data values $x_1, x_2, \ldots, x_n$ in ascending order by the similarity from the $F_{\text{text}}$. Then, each group should be formed by selecting elements such that each group contains the largest and smallest values available. Specifically, we can construct each group $G_i$ by taking the maximum and minimum values from the remaining elements.

**Proof.** To prove that these strategies effectively minimize $\sum_{i=1}^{k} \mu_i^2$, consider the following:

- **Assuming Constant Total:** Let's assume the overall sum of group means is constant, i.e., $\mu_1 + \mu_2 + \ldots + \mu_k = T$. The goal is to minimize $\sum_{i=1}^{k} \mu_i^2$ under this constraint.

- **Applying Cauchy-Schwarz Inequality:** By applying the Cauchy-Schwarz inequality in the context of these means:

$$k(\mu_1^2 + \mu_2^2 + \ldots + \mu_k^2) \geq (\mu_1 + \mu_2 + \ldots + \mu_k)^2 = T^2.$$

This implies that:

$$\mu_1^2 + \mu_2^2 + \ldots + \mu_k^2 \geq \frac{T^2}{k}.$$

Therefore, minimizing $\sum_{i=1}^{k} \mu_i^2$ occurs under the condition that the means are as equal as possible.

- **Validating the Construction Method:** Our group construction method ensures that all $\mu_i$ values are as equal as possible because we are taking elements from both ends of the distribution. This approach ensures that the means converge to the overall mean of the sample set, thereby fulfilling the necessary condition for minimizing $\sum_{i=1}^{k} \mu_i^2$.

- **Conclusion:** By focusing on strategies that leverage the largest and smallest available values for group construction and maintaining the overall sum of means as constant, we can effectively minimize the sum of squares of the means $\mu_1^2, \mu_2^2, \ldots, \mu_k^2$, thus maximizing the overall variance.

# B  Additional Experimental Analysis

## B.1  Detection Model Fine-tuning Necessity

To better demonstrate the necessity of the detection fine-tuning stage, we conduct comprehensive ablation experiments that demonstrate the effectiveness of our approach even without detection fine-tuning. As shown in Table 4, we evaluate our method without detection fine-tuning on TAO, using only our diffusion-based association enhancement. Our method still achieves competitive performance, particularly in association accuracy, indicating the core effectiveness of our diffusion-based approach.

Table 4: Performance comparison without detection fine-tuning on TAO validation set.

| Method | Novel | | | | Base | | | |
|---|---|---|---|---|---|---|---|---|
| | **TETA** | **LocA** | **AssoA** | **ClsA** | **TETA** | **LocA** | **AssoA** | **ClsA** |
| OVTrack | 27.8 | 48.8 | 33.6 | 1.5 | 35.5 | 49.3 | 36.9 | 20.2 |
| SLAck | 31.1 | 54.3 | 37.8 | 1.3 | 37.2 | 55.0 | 37.6 | 19.1 |
| OVTR | 31.4 | 54.4 | 34.5 | 5.4 | 36.6 | 52.2 | 37.6 | 20.1 |
| Ours (w/o det. fine-tune) | 32.4 | 55.3 | 39.9 | 2.1 | 38.7 | 55.3 | 41.1 | 19.8 |

Overall, our proposed method benefits from but is not dependent on detection fine-tuning. The complete version with fine-tuning constructs a comprehensive data-efficient training paradigm that maximizes the utility of limited sparse annotations, creating a holistic approach that leverages both association improvements (via diffusion) and detection improvements (via fine-tuning) to achieve optimal performance under data-constrained conditions.

## B.2  Analysis on Challenging Scenarios

To comprehensively assess our method's robustness, we systematically selected the most challenging videos from the TAO validation set (93 videos out of 988 total videos) using multiple quantitative criteria. Videos satisfying any of the following conditions were included in our challenging subset:

**Selection Criteria:**

- **Heavy Occlusion Scenarios:** Videos with substantial mutual occlusions where object bounding boxes exhibit IoU > 0.4 with other objects for more than 20% of the total frames.

- **Rapid Motion Patterns:** Videos with fast-moving objects where the average displacement ratio (bounding box center displacement to box diagonal length) > 0.3 across consecutive frames.

- **Frequent Entry/Exit Dynamics:** Videos where objects exhibit frequent entry/exit behaviors (more than 3 entry/exit cycles per trajectory on average).

- **High Object Density:** Videos containing crowded scenes with more than 8 concurrent objects per frame on average.

Using these systematic criteria, we identified videos presenting challenging tracking conditions, which we refer to as TAO-Hard. The evaluation results are shown in Table 5.
The results show different levels of performance degradation in these challenging scenarios, which is natural given the unpredictable object states, significant appearance changes, and objects leaving and re-entering the scene. Nevertheless, our method maintains superior performance compared to existing approaches, demonstrating strong robustness across diverse scenarios.

Table 5: Performance comparison on challenging scenarios (TAO-Hard).

| Method | Novel | | | | Base | | | |
|---|---|---|---|---|---|---|---|---|
| | TETA | LocA | AssoA | ClsA | TETA | LocA | AssoA | ClsA |
| OVTrack | 27.8 | 48.8 | 33.6 | 1.5 | 35.5 | 49.3 | 36.9 | 20.2 |
| OVTrack on TAO-Hard | 25.9(-1.9) | 45.3 | 31.3 | 1.1 | 33.9(-1.6) | 49.1 | 33.4 | 19.3 |
| OVTR on TAO | 31.4 | 54.4 | 34.5 | 5.4 | 36.6 | 52.2 | 37.6 | 20.1 |
| OVTR on TAO-Hard | 28.4(-2.0) | 51.2 | 32.1 | 1.9 | 35.2(-1.4) | 51.2 | 35.2 | 19.1 |
| Ours on TAO | 35.2 | 59.2 | 40.3 | 6.2 | 40.4 | 58.6 | 42.1 | 20.5 |
| Ours on TAO-Hard | 33.4(-1.8) | 57.6 | 39.2 | 3.4 | 39.2(-1.2) | 57.1 | 40.3 | 20.1 |

### B.3 Additional Association-Centric Metrics

To provide a more comprehensive evaluation of association performance, we include additional association-centric metrics as suggested by reviewers. Table 6 shows our experimental results on IDF1 and ID switches (IDSW) metrics.

Table 6: Association-centric metrics comparison on TAO validation set.

| Methods | IDF1 (%) | IDSW | MOTA (%) |
|---|---|---|---|
| OVTrack | 69.3 | 18,962 | 44.4 |
| OVTR | 72.6 | 12,128 | 48.2 |
| Ours | **76.2** | **10,393** | **57.3** |

Our method demonstrates strong association capability with the highest IDF1 (76.2%) and lowest ID switches (10,393), confirming the effectiveness of our diffusion-based approach for object association.

### B.4 Scalability Analysis

To analyze the computational efficiency with larger category sets, we conduct comprehensive experiments with class counts ranging from 10 to 1200 in Table 7. We measure the average time taken for dynamic group contrastive learning per batch, which includes both the construction of dynamic groups and the calculation of contrastive learning loss.

Table 7: Scalability analysis of dynamic group contrastive learning.

| Class counts | 10 | 100 | 200 | 400 | 800 | 1200 |
|---|---|---|---|---|---|---|
| Sec/batch | 0.11 | 0.13 | 0.15 | 0.20 | 0.24 | 0.27 |

The results demonstrate good scalability of the proposed method. Even at 1200 classes, the average time consumed per batch is only 0.27 seconds, with the majority of this time spent on dynamic group construction. The time required for contrastive learning loss calculation remains below 0.1 seconds across all class counts, demonstrating that our dynamic group contrastive learning approach scales efficiently and is not computationally prohibitive for very large category sets.

## C  Qualitative Analysis

We compare our method with the baseline method OVTrack across several challenging scenarios involving novel object classes. As shown in Figure 4, in the first construction site scene, our approach effectively and accurately tracks fast-moving drones, whereas OVTrack fails to detect the drones at all. Moreover, our method precisely classifies the bulldozer as a novel object category, while OVTrack misclassifies it as a truck and initially fails to detect the object entirely. Our method also demonstrates superior detection and tracking capabilities for base object classes (persons).

In the second racing scenario, characterized by high-speed vehicles and occlusion, our method successfully tracks and correctly classifies the race cars. In contrast, OVTrack struggles to detect occluded targets and exhibits incorrect ID switching. Its classification is also less precise, categorizing the vehicles under the broader "car" class instead of the specific "race car" category.

Figure 5 depicts a scene from the African savanna, featuring a novel category hippopotamus and two lions chasing it. Our method successfully classifies the hippopotamus correctly, whereas OVTrack misidentifies it as an "elephant". Additionally, OVTrack incorrectly labels the lion as a "horse" and

"cow". Moreover, our approach demonstrates superior detection and tracking accuracy compared to OVTrack.

Figure 6 illustrates the tracking performance in a field scenario, featuring a fast-moving dragonfly belonging to a novel category. Compared to OVTrack, our proposed method demonstrates superior detection and tracking capabilities, successfully identifying the dragonfly. In contrast, OVTrack fails to detect the dragonfly in most frames and cannot accurately classify it.

These results demonstrate that our method, through efficient training, significantly enhances localization, classification, and association capabilities across diverse and challenging tracking scenarios.

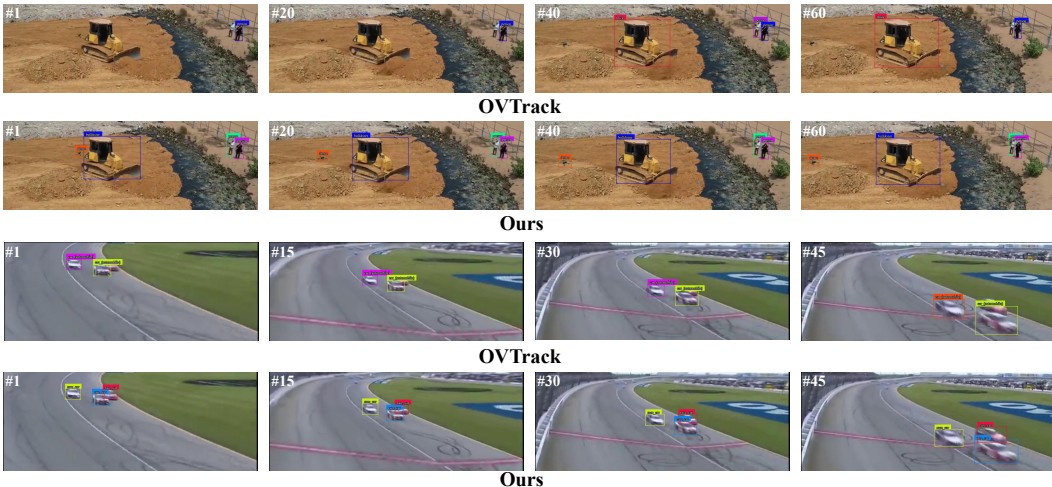

Figure 4: Visualization results from construction sites and race tracks with novel object categories, including drones and bulldozers in the construction scene, along with race cars in the racing track.

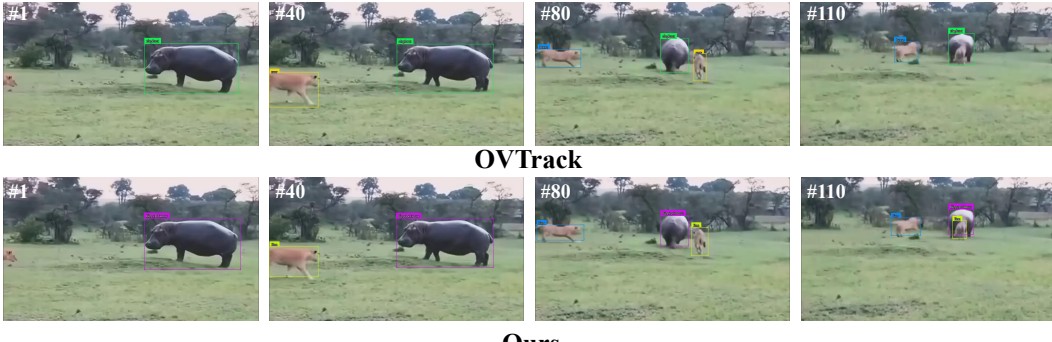

Figure 5: Visualization results in the African savanna with the novel object category, hippopotamus.

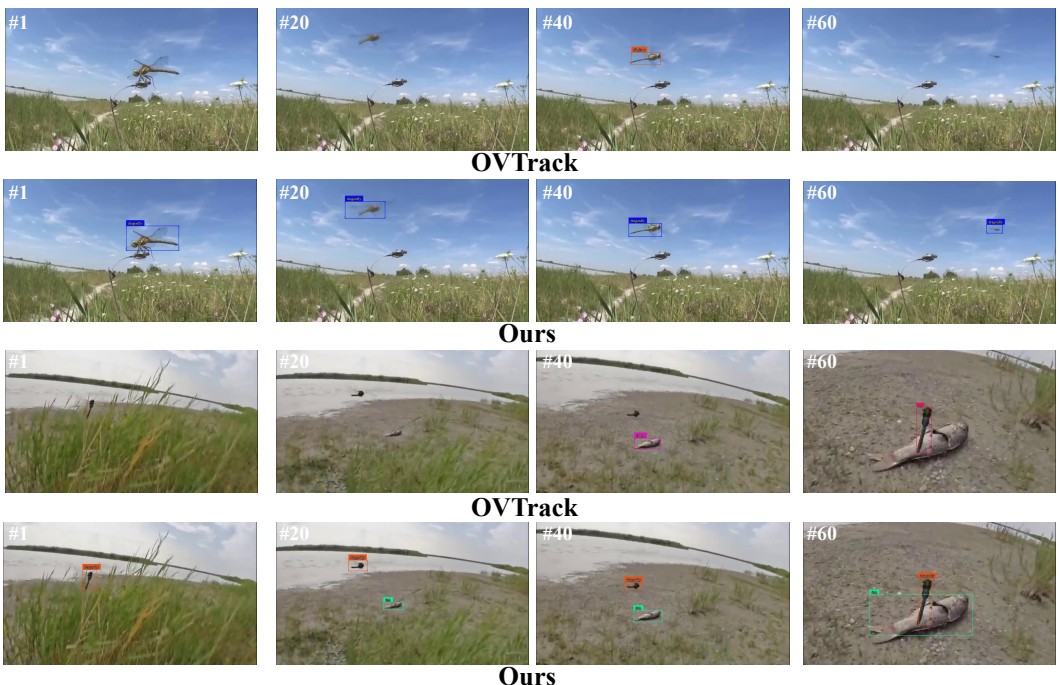

Figure 6: Visualization results in the field with the novel object category, dragonfly.

