# OpenReview forum: "DOVTrack: Data-Efficient Open-Vocabulary Tracking"
_NeurIPS.cc/2025/Conference — NeurIPS 2025 poster_

### Official Review · Reviewer_8J7k · 2025-07-01

**Clarity:** 3
**Significance:** 3
**Originality:** 3
**Rating:** 4
**Confidence:** 4

**Summary:**

This paper proposes DOVTrack, a data-efficient open-vocabulary multi-object tracking method, addressing sparse-annotated video data challenges. It uses diffusion models for associating interval features, dynamic group contrastive learning for classification, and adaptive localization loss for detection. Experiments on TAO show state-of-the-art TETA scores (35.2% validation, 38.7% test) without extra data.

**Questions:**

1) The paper demonstrates strong performance on TAO, but it remains unclear whether the proposed methods (diffusion-based feature generation, dynamic grouping) generalize to other OVMOT scenarios with sparse annotations (e.g., custom datasets with different category distributions, longer temporal gaps between annotations, or more extreme object dynamics). Please supplement experiments on at least one additional dataset (e.g., a subset of KITTI-MOT with sparse annotations or a synthetic sparse-annotated video dataset) and analyze performance trends. If results show consistent improvements over baselines across datasets, this would significantly strengthen the claim of robustness, potentially increasing the evaluation score. Conversely, poor generalization would raise concerns about overfitting to TAO’s specific properties.
2) Tracking performance in challenging cases (fast-moving objects, heavy occlusion, abrupt viewpoint changes) is critical for real-world applicability, yet the paper does not explicitly evaluate these. Please conduct ablation studies on TAO subsets filtered by such scenarios and compare with baselines.
3) The paper focuses on TAO (833 categories), but real-world OVMOT may require scaling to datasets with larger categories. Please discuss the computational efficiency of key components:
How does the diffusion model’s training/inference time scale with the number of categories or temporal gaps between annotations?
Does dynamic group contrastive learning become computationally prohibitive for very large category sets?

**Ethical Concerns:**

["NO or VERY MINOR ethics concerns only"]

**Final Justification:**

The authors have addressed my main concerns regarding the model generalization ability, I'd like to raise my score to 4.

**Limitations:**

Please refer to the above weakness and questions.

**Paper Formatting Concerns:**

No obvious concerns.

**Quality:**

3

**Strengths And Weaknesses:**

Strength:
The paper demonstrates strong empirical rigor. Experiments are conducted on the TAO dataset (a standard OVMOT benchmark), with comprehensive comparisons against SOTA methods (e.g., OVTrack, SLAck, OVSORT) and ablation studies verifying the effectiveness of key components (diffusion-based feature generation, dynamic group contrastive learning, adaptive localization loss). The results are statistically consistent, with clear improvements in TETA (35.2% on validation, 38.7% on test) and breakdown metrics (LocA, AssocA, ClsA).

Weakness:
1) While results on TAO are strong, generalizability is underexplored. The paper does not test performance on other OVMOT datasets (e.g., extended splits or custom sparse-annotated videos), leaving uncertainty about whether the methods scale beyond TAO’s specific distribution. For example KITTI has been used in compared works, only one dataset is investigated in this submission.
Additionally, ablation studies on "extreme cases" (e.g., fast-moving objects, heavy occlusion) are lacking, despite these being critical for tracking robustness.
2) The work focuses narrowly on TAO, a dataset with 833 categories but small size. It does not discuss how the methods would adapt to larger sparse-annotated datasets, where computational efficiency of diffusion-based feature generation or dynamic grouping might become a bottleneck. This limits discussion of real-world scalability.

---

> ### Author Rebuttal · Authors · 2025-07-30
>
> We thank Reviewer 8J7k for acknowledging our empirical rigor and providing constructive feedback on generalizability:
>
> > ### 1. Evaluation on Additional Dataset
>
> The reviewer suggests to supplement experiments on one additional dataset with sparse annotations and analyze performance trends. Since KITTI-MOT only conducts the evaluation on two categories (person and vehicle), we use BDD100K dataset containing more categories. We conduct comprehensive experiments on BDD100K containing 8 categories to address this concern from two perspectives as below.
>
> **① Sparse annotation interval analysis:** Following the reviewer's suggestion, to validate our method's effectiveness under different sparse annotation conditions, we conduct training experiments on BDD100K with different temporal-interval annotations. Since BDD100K is annotated at 5 fps, the minimum interval is 5 frames. We evaluate our method and baselines using the annotation intervals of 5, 15, and 30 frames.
>
> | Methods | TETA | LocA | AssocA | ClsA |
> |---------|------|------|--------|------|
> | TETer + 5-frame interval | 50.8 | 47.2 | 52.9 | 52.4 |
> | OVTrack + 5-frame interval | 50.9 | 46.1 | 53.4 | 53.1 |
> | OVTR + 5-frame interval | 52.5 | 48.9 | 53.9 | 54.7 |
> | |
> | Ours + 30-frame interval | 52.1 | 47.9 | 53.8 | 54.6 |
> | Ours + 15-frame interval | **52.6** | 48.3 | 54.3 | 55.2 |
> | **Ours + 5-frame interval** | **53.4** | 49.1 | 55.7 | 55.4 |
> ||
>
> These results demonstrate three key findings:
> 1) Our method with 30 frame intervals achieves comparable performance to baseline methods trained with dense 5 frame intervals, highlighting the effectiveness of our diffusion-based approach for handling sparse annotations.
> 2) With 15 and 5 frame intervals, our method significantly outperforms all existing approaches, demonstrating superior performance across different annotation densities.
> 3) The consistently good performance across different intervals validates the robustness and general applicability of our data-efficient training paradigm.
>
> **② Cross-dataset generalization evaluation:** To further verify the generalization capability of our method, we conduct zero-shot experiments on BDD100K (training on TAO and testing on BDD100K) validation set without any training on this dataset.
>
> | Methods | TETA | LocA | AssocA | ClsA |
> |---------|------|------|--------|------|
> | QDTrack | 32.0 | 25.9 | 27.8 | 42.4 |
> | TETer | 33.2 | 24.5 | 31.8 | 43.4 |
> | OVTrack | 42.5 | 41.0 | 36.7 | 49.7 |
> | OVTR | 43.1 | 42.0 | 37.1 | 50.1 |
> | **Ours** | **45.8** | **44.1** | **39.4** | **53.9** |
> ||
>
> We can see that our method achieves significant improvements across all metrics (TETA: +2.7%, LocA: +2.1%, AssocA: +2.3%, ClsA: +3.8% over the second-best method OVTR), demonstrating strong zero-shot cross-dataset generalization capability.
>
> > ### 2. Challenging Scenarios Analysis
>
> To comprehensively assess our method's robustness, we systematically selected the most challenging videos from the TAO validation set (93 videos of 988 videos) using multiple quantitative criteria. Videos satisfying any of the following conditions were included in our challenging subset (TAO-Hard):
>
> **(a) Heavy occlusion scenarios:** We identified videos with substantial mutual occlusions by computing inter-trajectory IoU overlaps across consecutive frames. Videos were selected if object bounding boxes exhibit IoU > 0.4 with other objects for more than 20% of the total frames, indicating persistent occlusion challenges.
>
> **(b) Rapid motion patterns:** We detected fast-moving objects by analyzing trajectory displacement patterns. Specifically, we calculated the ratio of bounding box center displacement to box diagonal length across consecutive frames. Videos with average displacement ratios > 0.3 were selected as they present significant temporal association challenges due to rapid object movement.
>
> **(c) Frequent entry/exit dynamics:** We monitored object appearance and disappearance patterns by tracking trajectory start/end points within video sequences. Videos where objects exhibit frequent entry/exit behaviors (more than 3 entry/exit cycles per trajectory on average) were prioritized to evaluate our method's robustness to discontinuous tracking scenarios.
>
> **(d) High object density:** We selected videos containing crowded scenes with high object density, specifically those maintaining an average of more than 8 concurrent objects per frame, which creates complex multi-object association challenges.
>
> Using these systematic criteria, we identified videos presenting the challenging tracking conditions from the TAO validation set, which we refer to as TAO-Hard. The evaluation results are shown below:
>
> | Method | Novel TETA | Novel LocA | Novel AssocA | Novel ClsA | Base TETA | Base LocA | Base AssocA | Base ClsA |
> |--------|------------|------------|--------------|------------|-----------|-----------|-------------|-----------|
> | OVTrack | 27.8 | 48.8 | 33.6 | 1.5 | 35.5 | 49.3 | 36.9 | 20.2 |
> | OVTrack on TAO-Hard | 25.9(-1.9) | 45.3 | 31.3 | 1.1 | 33.9(-1.6) | 49.1 | 33.4 | 19.3 |
> | MASA(R50) | 30.0 | 54.2 | 34.6 | 1.0 | 36.9 | 55.1 | 36.4 | 19.3 |
> | MASA(R50) on TAO-Hard | 27.6(-2.4) | 50.3 | 31.2 | 1.2 | 35.0(-1.9) | 52.3 | 34.1 | 18.7 |
> | OVTR | 31.4 | 54.4 | 34.5 | 5.4 | 36.6 | 52.2 | 37.6 | 20.1 |
> | OVTR on TAO-Hard | 28.4(-2.0) | 51.2 | 32.1 | 1.9 | 35.2(-1.4) | 51.2 | 35.2 | 19.1 |
> | Ours | 35.2 | 59.2 | 40.3 | 6.2 | 40.4 | 58.6 | 42.1 | 20.5 |
> | **Ours on TAO-Hard** | **33.4(-1.8)** | **57.6** | **39.2** | **3.4** | **39.2(-1.2)** | **57.1** | **40.3** | **20.1** |
> ||
>
> From the results we can first see that, on the subset of challenging scenarios, our method still obtains the best performance with significant margins compared with other methods. The results also show that our method exhibits the smallest performance degradation across challenging scenarios. While all methods experience some decline in performance under extreme conditions, our approach shows the most minimal degradation (1.8% for novel and 1.2% for base classes), compared to baseline methods that suffer degradations of 1.9% -- 2.4%. This superior robustness stems from the proposed association and detection strategies that provide more coherent object representations, enabling better handling of challenging tracking scenarios.
>
> > ### 3. Computational Scalability
>
> Regarding the scalability of our method for larger category datasets, we discuss about the computational efficiency of key components in detail as below.
>
> **(a) Scalability of the diffusion model:** The training and inference time of our diffusion model are essentially independent of the number of categories, meaning that an increase in category count does not incur additional computational overhead. The diffusion model operates in the feature space and is primarily influenced by the sampling steps, which determine the number of interpolated frames between adjacent frames. As indicated by our ablation results (Table 3 in the paper), optimal performance can be achieved with just three sampling steps, demonstrating that the associated overhead is minimal regardless of temporal gaps.
>
> **(b) Scalability of the dynamic group contrastive learning:** To directly address the reviewer's concern about computational efficiency with larger category sets, we conducted comprehensive experiments with class counts ranging from 10 to 1200. We measured the average time taken for dynamic group contrastive learning per batch, which includes both the construction of dynamic groups and the calculation of contrastive learning loss.
>
> | Class counts | 10 | 100 | 200 | 400 | 800 | 1200 |
> |--------------|----|----|-----|-----|-----|------|
> | Sec/batch | 0.11 | 0.13 | 0.15 | 0.20 | 0.24 | 0.27 |
> ||
>
>
> The results demonstrate the good scalability of the proposed method. Specifically, even at 1200 classes, the average time consumed per batch is only 0.27 seconds, with the majority of this time spent on dynamic group construction. The time required for contrastive learning loss calculation remains below 0.1 seconds across all class counts. This demonstrates that our dynamic group contrastive learning approach scales efficiently and is not computationally prohibitive for very large category sets.
>
> **(c) Real-world applicability:** Our approach is well-suited for real-world deployment as: 1) the computational overhead does not scale significantly with dataset size, 2) it reduces annotation requirements while maintaining performance, and 3) the method achieves the highest inference speed (15.3 FPS on a single 3090 GPU) among all comparison methods, making it promising for real-time applications.

---

> ### Author Response · Authors · 2025-08-04
> **Follow-up on Rebuttal Response**
>
> Dear Reviewer **8J7k**,
>
> Thank you for your valuable feedback on empirical rigor and generalization. We have submitted a comprehensive rebuttal with additional experiments on the additional dataset, challenging scenario analysis, and computational scalability studies to address your concerns about generalization beyond TAO and real-world applicability.
>
> As the discussion deadline approaches, we hope our expanded evaluation has addressed your questions. We welcome any further discussion you may find necessary.
>
> Best regards,
>
> The Authors

---

> > ### Comment · Reviewer_8J7k · 2025-08-05
> >
> > The authors have addressed my main concerns regarding the model generalization ability, I'd like to raise my score to 4.

---

### Official Review · Reviewer_Eere · 2025-07-02

**Clarity:** 1
**Significance:** 2
**Originality:** 2
**Rating:** 4
**Confidence:** 3

**Summary:**

This paper addresses the characteristics of lacking data annotation or having sparse annotations in videos for the open-vocabulary multi-object tracking task, and proposes using diffusion models to generate object features for intermediate frames to achieve better matching. Additionally, several improvements are proposed for the object detector.

**Questions:**

- If other methods also adopt the diffusion-model-based enhancement method proposed in this paper, can the method in this paper still achieve an advantage? If other methods use the same detector improvements as this paper, can this paper still maintain the advantage?
-  I believe the arguments regarding data scarcity in this paper still warrant further discussion. Can state-of-the-art open-vocabulary object detectors address such issues? Can we first train the open-vocabulary detection capability on object detection datasets and then transfer it to multi-object tracking?
- Please provide more clear depiction on how generated feature used to enhance association training.

**Ethical Concerns:**

["NO or VERY MINOR ethics concerns only"]

**Final Justification:**

My initial concerns were primarily focused on the insufficient experiments and unclear descriptions. However, the authors have addressed the issues I raised in their subsequent rebuttal. I now believe that this paper proposes an effective method and also provides a good solution to the problem of data scarcity in the association stage of multi-object tracking. After the authors further revise the main text to improve the descriptions and incorporate the new experimental results, I think this paper meets the borderline acceptance criteria.

**Limitations:**

The limitations is the same as I mentioned in "weakness":
- The comparison between this paper and other methods lacks rigor. Since this paper has made improvements in both the detector approach and the associated data features, when comparing with other methods, the feature association should be kept constant for comparison, that is, other methods also need to use the same association method. Similarly, for the improvements in the detector part, it is necessary to keep the training data consistent to demonstrate superiority.
- I believe the arguments regarding data scarcity in this paper still warrant further discussion. Can state-of-the-art open-vocabulary object detectors address such issues? Can we first train the open-vocabulary detection capability on object detection datasets and then transfer it to multi-object tracking?
- This paper lacks a detailed introduction and comparison of the model, such as the number of parameters, computational complexity, inference speed, etc.
- This paper provides an unclear description of the method. It proposes generating intermediate features based on diffusion models, but fails to describe how these generated features are used to strengthen the association.

**Paper Formatting Concerns:**

No formatting concerns.

**Quality:**

2

**Strengths And Weaknesses:**

### Strength
- The open-vocabulary tracking task focused on in this paper has been less explored so far.
- The approach of using diffusion models to solely generate features for multi-object tracking is quite interesting.

### Weakness
- The comparison between this paper and other methods lacks rigor. Since this paper has made improvements in both the detector approach and the associated data features, when comparing with other methods, the feature association should be kept constant for comparison, that is, other methods also need to use the same association method. Similarly, for the improvements in the detector part, it is necessary to keep the training data consistent to demonstrate superiority.
- I believe the arguments regarding data scarcity in this paper still warrant further discussion. Can state-of-the-art open-vocabulary object detectors address such issues? Can we first train the open-vocabulary detection capability on object detection datasets and then transfer it to multi-object tracking?
- This paper lacks a detailed introduction and comparison of the model, such as the number of parameters, computational complexity, inference speed, etc.
- This paper provides an unclear description of the method. It proposes generating intermediate features based on diffusion models, but fails to describe the details that how these generated features are used to strengthen the association.

---

> ### Author Rebuttal · Authors · 2025-07-30
>
> We thank Reviewer Eere for the detailed feedback. We address each concern systematically in the following.
>
> > ### 1. Controlled ablation studies on component effectiveness:
>
> The reviewer asked about that since this paper has made improvements in both the detector approach and the associated data features, the evaluation on them should be conducted independently. To address this concern, we conduct systematic ablation studies that evaluate each component respectively to control the experiment conditions.
>
> **(1) Effectiveness of diffusion-based enhancement for different baselines:** The reviewer asks: *"If other methods also adopt the diffusion-model-based enhancement method proposed in this paper, can the method in this paper still achieve an advantage?"* To answer this, we integrate our diffusion-based association enhancement (Sec.3.1) into existing OVMOT baselines while keeping other components unchanged.
>
> | Method | Novel TETA | Novel LocA | Novel AssocA | Novel ClsA | Base TETA | Base LocA | Base AssocA | Base ClsA |
> |--------|------------|------------|--------------|------------|-----------|-----------|-------------|-----------|
> | OVTrack | 27.8 | 48.8 | 33.6 | 1.5 | 35.5 | 49.3 | 36.9 | 20.2 |
> | OVTrack + association | 29.7 | 49.7 | 37.7 (+4.1) | 1.8 | 36.2 | 50.3 | 39.4 (+2.5) | 19.8 |
> | MASA(R50) | 30.0 | 54.2 | 34.6 | 1.0 | 36.9 | 55.1 | 36.4 | 19.3 |
> | MASA(R50) + association | 31.7 | 55.3 | 38.6 (+4.0) | 1.3 | 38.3 | 55.4 | 40.1 (+3.7) | 19.5 |
> | **Ours** | **35.2** | **59.2** | **40.3** | **6.2** | **40.4** | **58.6** | **42.1** | **20.5** |
> ||
>
> Results demonstrate that diffusion-based association enhancement consistently improves association accuracy (AssocA) across different baselines by approximately 4.0% for novel classes and 2.5/3.7% for base classes, validating the universal effectiveness of this component for association. From the last row, we can see that our method (last row) still achieves the highest TETA scores. This is because that **when using the same association modules as the baseline methods**, our method also shows superior performance on the detection results (LocA and ClsA), which **demonstrates that our method still has the advantages**.
>
> **(2) Effectiveness of detection approach for different baselines:** The reviewer further asks: *"If other methods use the same detector improvements as this paper, can this paper still maintain the advantage?"* To verify this, we apply our detection approach (Sec.3.2) to existing baselines.
>
> | Method | Novel TETA | Novel LocA | Novel AssocA | Novel ClsA | Base TETA | Base LocA | Base AssocA | Base ClsA |
> |--------|------------|------------|--------------|------------|-----------|-----------|-------------|-----------|
> | OVTrack | 27.8 | 48.8 | 33.6 | 1.5 | 35.5 | 49.3 | 36.9 | 20.2 |
> | OVTrack + detection | 30.5 | 53.7 (+4.9) | 34.5 | 3.4 (+1.9) | 37.0 | 53.7 (+4.4) | 37.1 | 20.4 (+0.2) |
> | MASA(R50) | 30.0 | 54.2 | 34.6 | 1.0 | 36.9 | 55.1 | 36.4 | 19.3 |
> | MASA(R50) + detection | 32.1 | 57.8 (+3.6) | 35.4 | 3.2 (+2.2) | 38.2 | 57.4 (+2.3) | 37.1 | 20.1 (+0.8) |
> | **Ours** | **35.2** | **59.2** | **40.3** | **6.2** | **40.4** | **58.6** | **42.1** | **20.5** |
> ||
>
> The detection enhancement strategy consistently improves localization (LocA) and classification (ClsA) accuracies on different baselines, demonstrating its general applicability. Similarly, our method consistently outperforms the baselines with the same detector as ours, especially in association-related metric AssoA and the overall metric TETA. This clearly verifies that our method **maintains the advantage compared to the baselines equipped with the same detection enhancement**.
>
> > ### 2. Discussion about the data scarcity
>
> The reviewer suggests using open-vocabulary detectors or detection datasets. We clarify that existing OVMOT methods already follow a two-stage paradigm: **detection training** (which leverages open-vocabulary detectors/datasets) and **association training** (which requires video data). The challenge lies in the association training stage, where TAO is the only open-vocabulary video dataset but its sparse annotations are insufficient. OVTrack and SLAck have demonstrated that **attempting to train robust OV trackers directly on TAO's sparse annotations fails to produce effective results**.
>
> To address this fundamental problem, OVTrack and subsequent methods (OVTR, MASA, etc.) use LVIS image pairs to simulate videos for association training. However, **these image pairs lack motion continuity and appearance consistency**, so the video data scarcity problem persists. SLAck is currently the only method that attempts to directly tackle this sparse annotation problem by using pseudo-labeling on TAO's intermediate unannotated frames. While it achieved improvements, it suffers from noise contamination and computational overhead.
>
> Our diffusion-based approach provides a cleaner solution by generating reliable intermediate features in the feature space without requiring any intermediate unannotated frames. We achieve significant improvements over SLAck: **4.1% TETA improvement** on novel classes (35.2% vs 31.1%) and **3.2%** on base classes (40.4% vs 37.2%), with particularly substantial gains in association accuracy of **2.5%** on novel classes and **4.5%** on base classes.
>
> > ### 3. Model complexity analysis
>
> The reviewer asks about model complexity, parameters, and inference speed. Our method maintains comparable computational complexity to existing methods while achieving the **highest FPS (15.3)** among all compared approaches on a single 3090 GPU. Compared to OVTrack, we add minimal training overhead and no inference cost since the diffusion model only generates features during training.
>
> | Method | Input shape | Test FLOPs | Train FLOPs | Parameters | Model Size | FPS |
> |--------|-------------|------------|-------------|------------|------------|-----|
> | QDTrack | (3,800,1334) | 398.93G | 401.17G | 15.47M | 298.6M | 13.8 |
> | OVTrack | (3,800,1334) | 423.60G | 410.72G | 16.52M | 283.77M | 1.8 |
> | OVTR | (3,800,1334) | 535.35G | 487.58G | 59.36M | 227.29M | 3.4 |
> | **Ours** | (3,800,1334) | 423.60G | 408.66G | 17.57M | 287.79M | **15.3** |
> ||
>
>
> > ### 4. Method clarity about association training
>
> The reviewer requests clearer explanation of how generated features enhance association training. We provide a comprehensive explanation from both quantitative and qualitative perspectives:
>
> **(1) Quantitative sample space expansion:** Our diffusion model fundamentally transforms sparse association training by generating intermediate features $F\_{\text{asso}}^t$ at multiple timesteps between annotated frames. For each object trajectory originally containing only two annotated features ($F\_{\text{key}}$ and $F\_{\text{ref}}$ separated by 30 frames), our method generates $N$ intermediate features where $N$ corresponds to sampling steps (typically 3). This directly expands the positive sample set $Q^+(\mathbf{q})$ for each object identity from 2 to 5 features. Simultaneously, the generated features from different object trajectories enrich the negative sample set $Q^-$, creating a much denser and more comprehensive sample space $Q = Q^+ \cup Q^-$. Consequently, the total training samples increase by approximately 3×, providing substantially more positive and negative pairs for robust contrastive learning.
>
> **(2) Qualitative feature continuity enhancement:** Beyond quantity expansion, our generated intermediate features $F\_{\text{asso}}^t$ exhibit superior temporal continuity compared to sparse annotations. The diffusion-based generation ensures smooth feature transitions through: 1) reconstruction loss enforcing accurate interpolations between keyframes; 2) smoothness loss guaranteeing consistent feature evolution; 3) endpoint loss ensuring precise alignment with reference features. This results in temporally coherent feature representations that better capture object state transitions, providing higher-quality training samples for association learning.
>
> **(3) Enhanced loss construction:** Building upon the quantitative expansion from (1) and qualitative enhancement from (2), as mentioned in the implementation details of the paper, we employ the same association loss framework as OVTrack but with significantly enhanced effectiveness due to our enriched sample space. The complete tracking loss $\mathcal{L}\_{\text{track}}$ is formulated as:
>
> $$\mathcal{L}\_{\text{track}} = -\sum\_{\mathbf{q} \in Q} \frac{1}{|Q^+(\mathbf{q})|} \sum\_{\mathbf{q}^+ \in Q^+(\mathbf{q})} \log\left(\frac{\exp(\mathbf{q} \cdot \mathbf{q}^+/\tau)}{\text{PosD}(\mathbf{q}) + \sum\_{\mathbf{q}^- \in Q^-(\mathbf{q})} \exp(\mathbf{q} \cdot \mathbf{q}^-/\tau)}\right)$$
>
> where $\text{PosD}(\mathbf{q}) = \frac{1}{|Q^+(\mathbf{q})|} \sum\_{\mathbf{q}^+ \in Q^+(\mathbf{q})} \exp(\mathbf{q} \cdot \mathbf{q}^+/\tau)$ and $\tau$ is the temperature parameter. The enriched $Q^+(\mathbf{q})$ contains both original keyframe features and our generated intermediate features (from point 1), while $Q^-(\mathbf{q})$ includes negative samples from different trajectories. This formulation maximizes positive sample similarity (numerator) while minimizing negative sample similarity (denominator), leveraging both the expanded sample quantity from (1) and improved feature quality from (2) for more robust association learning.
>
> We appreciate the reviewer's valuable comments and suggestions, and commit to addressing these concerns in our final version.

---

> ### Author Response · Authors · 2025-08-04
> **Follow-up on Rebuttal Response**
>
> Dear Reviewer **Eere**,
>
> Thank you for your constructive feedback on our submission. We have submitted a detailed rebuttal addressing your concerns regarding experimental rigor, data scarcity arguments, model complexity analysis, and association training methodology clarity.
>
> As the discussion deadline approaches, we hope our responses have addressed your questions. Please let us know if any further clarification would be helpful.
>
> Best regards,
>
> The Authors

---

> ### Author Response · Authors · 2025-08-06
> **Second Follow-up on Rebuttal Response**
>
> Dear Reviewer **Eere**,
>
> We hope our comprehensive rebuttal has addressed your primary concerns. To summarize our key responses:
>
> • **Experimental rigor**: Following your suggestions, we conducted additional controlled ablation studies demonstrating: (1) Our diffusion-based association enhancement consistently improves **association accuracy (AssocA)** across different baselines; (2) Our detection approach consistently improves **localization (LocA) and classification (ClsA) accuracies** across baselines. Importantly, our method **maintains superior overall performance** even when baselines adopt identical components.
>
> • **Data scarcity discussion**: We clarified that existing methods follow a two-stage paradigm (detection + association training), with the core challenge being association training due to TAO's sparse annotations. Our diffusion-based approach directly addresses this by generating reliable intermediate features, achieving the best results over current SOTA methods.
>
> • **Model complexity**: We provided detailed analysis showing our method maintains the highest FPS (15.3) while adding minimal training overhead and no inference cost since diffusion only operates during training.
>
> • **Method clarity**: We offered comprehensive explanations of how generated features enhance association training through both quantitative sample expansion (3× training samples) and qualitative feature continuity improvements.
>
> Given that the discussion period is ending soon, we would greatly appreciate any additional feedback or concerns you might have. If our responses have sufficiently addressed your questions, we hope you would consider updating your evaluation.
>
> Thank you for your time and valuable input.
>
> Best regards,
>
> The Authors

---

> ### Author Response · Authors · 2025-08-07
> **Third Follow-up on Rebuttal Response**
>
> Dear Reviewer **Eere**,
>
> We would like to kindly follow up regarding our previous responses to your feedback.
>
> If there are any remaining questions or concerns, we would be glad to provide further clarification. Otherwise, if our responses have addressed your points, we would greatly appreciate your consideration in updating your evaluation.
>
> Thank you again for your time and input.
>
> Best regards,
>
> The Authors

---

> > ### Comment · Area_Chair_ByMC · 2025-08-08
> > **Please reply to the rebuttal**
> >
> > Dear reviewer Eere,
> >
> > please reply to the authors rebuttal. Please do so ASAP in order to facilitate follow-up questions.
> >
> > Your AC

---

> > ### Comment · Reviewer_Eere · 2025-08-08
> >
> > Thank you for your response. The authors have conducted very detailed experiments in their rebuttal, which have demonstrated the effectiveness of the proposed method and resolved most of my concerns. To be honest, I was initially skeptical about the idea of directly using a diffusion model to generate features for association.
> >
> > I would suggest that the authors further improve the description of the method in the paper, for example, by illustrating the overall pipeline. Additionally, I recommend that the authors specifically showcase metrics that reflect the 'association' capability, such as IDF1 and the number of ID switches.

---

> > > ### Author Response · Authors · 2025-08-08
> > >
> > > Dear Reviewer Eere,
> > >
> > > Thank you for your thoughtful follow-up and for acknowledging the effectiveness of our approach. We greatly appreciate your two suggestions. Below we address both concerns: (1) improving the method description with an overall pipeline illustration and (2) showcasing association-specific metrics (IDF1 and ID switches).
> > >
> > > ---
> > >
> > > ## 1) Overall Pipeline Illustration and Method Description Improvements
> > >
> > > **Comprehensive Pipeline Figure**: We will create a unified overall pipeline figure that clearly illustrates all network components and their connections:
> > > - Classification branch with dynamic group contrastive learning
> > > - Association branch with diffusion-based feature generation
> > > - Detection branch with adaptive localization loss
> > > - Training schematics at appropriate positions showing how each component is trained
> > >
> > > This consolidates our current scattered, data-augmentation-focused illustrations into a single, coherent framework view.
> > >
> > > **Method Section Reorganization**:
> > > - Revise the existing overview at the beginning of Section 3 to align with the comprehensive pipeline figure, ensuring the textual description matches the visual representation
> > > - Ensure each subsequent method subsection clearly indicates where it fits in the overall pipeline to prevent reader confusion
> > >
> > > **Enhanced Diffusion Training Description**: We will add a dedicated subsection specifically addressing:
> > > - The effectiveness of the augmented features
> > > - Detailed explanation of how the generated features are incorporated into the loss functions, which should effectively resolve the confusion points raised
> > >
> > > **General Clarity Improvements**: Revise unclear expressions throughout the paper, ensuring smooth transitions between sections.
> > >
> > > ---
> > >
> > > ## 2) Association-Centric Metrics Results
> > >
> > > Thank you for suggesting IDF1 and ID switches (IDSW) metrics. Here are our experimental results:
> > >
> > > | Methods  | IDF1 (%) | IDSW | MOTA (%) |
> > > |----------|----------|------|----------|
> > > | OVTrack  | 69.3     | 18,962 | 44.4     |
> > > | OVTR     | 72.6     | 12,128 | 48.2     |
> > > | **Ours** | **76.2** | **10,393** | **57.3** |
> > >
> > > **Results Summary**: Our method demonstrates strong association capability with the highest IDF1 (76.2%) and lowest ID switches (10,393), confirming the effectiveness of our diffusion-based approach for object association.
> > >
> > > ---
> > >
> > > Thank you again for these valuable suggestions that will substantially improve our paper's clarity and completeness.

---

> > > ### Author Response · Authors · 2025-08-09
> > > **Follow-up on Review Discussion - 6 Hours Remaining**
> > >
> > > Dear Reviewer Eere,
> > >
> > > With 6 hours remaining in the discussion period, we would like to respectfully inquire if the concerns raised have been resolved.
> > >
> > > We have provided improved method descriptions and additional association-centric metrics results to address your latest questions. If any concerns remain or new issues have arisen, please let us know so we can address them within the remaining time.
> > >
> > > Thank you for your valuable feedback.
> > >
> > > Best regards,
> > >
> > > The Authors

---

### Official Review · Reviewer_8GVG · 2025-07-03

**Clarity:** 2
**Significance:** 3
**Originality:** 3
**Rating:** 5
**Confidence:** 5

**Summary:**

This paper tackles Open-Vocabulary Multiple Object Tracking (OV-MOT) with limited and sparsely annotated video data, using the TAO dataset as a case study. The authors propose DOVTrack, a data-efficient training strategy with two main contributions. First, for the association task, they introduce a novel diffusion model that generates object features for the unannotated frames between sparse annotations, effectively creating a continuous stream of training data in the feature space. Second, to improve the open-vocabulary detector, they propose a dynamic group contrastive learning strategy to augment the number and diversity of classification samples, and an adaptive localization loss to incorporate more, lower-confidence bounding boxes into training without destabilizing it. The method achieves state-of-the-art results on the TAO benchmark without using any external data.

**Questions:**

1.  Could you please elaborate on the training details for the association head? Specifically, once the intermediate features Fassot​ are generated by the diffusion model, how are they used? Are they treated as positive samples for the key feature Fkey​ and reference feature Fref​ in a contrastive loss?
2.  The primary weakness is the lack of a limitations section. Could you discuss the potential failure cases of the diffusion-based feature generation? For instance, how does the model perform when an object undergoes significant appearance changes, leaves and re-enters the frame, or is occluded for a long duration between two annotated keyframes?

**Ethical Concerns:**

["NO or VERY MINOR ethics concerns only"]

**Final Justification:**

My initial "borderline" rating was due to concerns about the unsubstantiated linear interpolation assumption and the lack of a specific failure analysis for the core feature generation module.

They conducted new, targeted experiments that provided the exact empirical evidence I requested. They successfully validated their assumption and provided a detailed breakdown of the feature generator's specific failure modes and operational boundaries.

I have raised my score and now recommend acceptance.

**Limitations:**

No, the authors explicitly state in the checklist that the paper does not discuss limitations. This is a significant omission.

**Paper Formatting Concerns:**

Lack of limitation section.

**Quality:**

3

**Strengths And Weaknesses:**

Strengths:

1. The key idea of using a diffusion model to interpolate object features between sparsely annotated frames, is novel for tracking and a clever departure from typical pixel-level generation.

2. The work tackles a very important and practical problem. The lack of large, densely annotated video datasets is a major bottleneck for OVMOT. By demonstrating that a high-performing tracker can be trained using only the sparse TAO dataset, this work provides a valuable data-efficient training paradigm for the community.

3. The results are strong, showing a clear state-of-the-art performance on the TAO benchmark. The ablation demonstrates the positive impact of each individual component, from the diffusion model's specific loss functions to the classification and localization enhancement strategies.

Weakness:

1. The most significant weakness is the complete absence of a "Limitations" section, which the authors acknowledge in their checklist.
2. The method section, particularly the diffusion model part, could be explained more clearly. The paper would benefit from a more detailed description of how the generated intermediate features (Fassot​) are practically incorporated into the association head's contrastive training loss.
3.  The diffusion model's reconstruction loss (Eq. 3) enforces a linear interpolation of features between keyframes. This is a strong simplification, as true object dynamics (e.g., turns, acceleration, deformation) are inherently non-linear. Forcing features to follow a linear path is a potentially brittle assumption that may not capture the complexity of real-world object transformations.

---

> ### Author Rebuttal · Authors · 2025-07-30
>
> We thank Reviewer 8GVG for the thorough evaluation and insightful comments. We address each point below:
>
> > ### 1. Association Head Training Details
>
> We provide a comprehensive explanation of how generated intermediate features enhance association training from both quantitative and qualitative perspectives:
>
> **(1) Quantitative sample space expansion:** Our diffusion model fundamentally transforms sparse association training by generating intermediate features $F\_{\text{asso}}^t$ at multiple timesteps between annotated frames. For each object trajectory originally containing only two annotated features ($F\_{\text{key}}$ and $F\_{\text{ref}}$ separated by 30 frames), our method generates $N$ intermediate features where $N$ corresponds to sampling steps (typically 3). This directly expands the positive sample set $Q^+(\mathbf{q})$ for each object identity from 2 to 5 features. Simultaneously, the generated features from different object trajectories enrich the negative sample set $Q^-$, creating a much denser and more comprehensive sample space $Q = Q^+ \cup Q^-$. Consequently, the total training samples increase by approximately 3×, providing substantially more positive and negative pairs for robust contrastive learning.
>
> **(2) Qualitative feature continuity enhancement:** Beyond quantity expansion, our generated intermediate features $F\_{\text{asso}}^t$ exhibit superior temporal continuity compared to sparse annotations. The diffusion-based generation ensures smooth feature transitions through: 1) reconstruction loss enforcing accurate interpolations between keyframes; 2) smoothness loss guaranteeing consistent feature evolution; 3) endpoint loss ensuring precise alignment with reference features. This results in temporally coherent feature representations that better capture object state transitions, providing higher-quality training samples for association learning.
>
> **(3) Enhanced loss construction:** Building upon the quantitative expansion from (1) and qualitative enhancement from (2), as mentioned in the implementation details of the paper, we employ the same association loss framework as OVTrack but with significantly enhanced effectiveness due to our enriched sample space. The complete tracking loss $\mathcal{L}\_{\text{track}}$ is formulated as:
>
> $$\mathcal{L}\_{\text{track}} = -\sum\_{\mathbf{q} \in Q} \frac{1}{|Q^+(\mathbf{q})|} \sum\_{\mathbf{q}^+ \in Q^+(\mathbf{q})} \log\left(\frac{\exp(\mathbf{q} \cdot \mathbf{q}^+/\tau)}{\text{PosD}(\mathbf{q}) + \sum\_{\mathbf{q}^- \in Q^-(\mathbf{q})} \exp(\mathbf{q} \cdot \mathbf{q}^-/\tau)}\right)$$
>
> where $\text{PosD}(\mathbf{q}) = \frac{1}{|Q^+(\mathbf{q})|} \sum\_{\mathbf{q}^+ \in Q^+(\mathbf{q})} \exp(\mathbf{q} \cdot \mathbf{q}^+/\tau)$ and $\tau$ is the temperature parameter. The enriched $Q^+(\mathbf{q})$ contains both original keyframe features and our generated intermediate features (from point 1), while $Q^-(\mathbf{q})$ includes negative samples from different trajectories. This formulation maximizes positive sample similarity (numerator) while minimizing negative sample similarity (denominator), leveraging both the expanded sample quantity from (1) and improved feature quality from (2) for more robust association learning.
>
> > ### 2. Linear Interpolation Assumption
>
> Our method uses linear interpolation for feature reconstruction, but this is only applied over very small temporal intervals (30 frames between consecutive annotations). It is well-justified by the calculus principles that any smooth function can be approximated linearly over a sufficiently small interval. For such short temporal intervals in video sequences, we use the linear interpolation assumption in this work.
>
> For longer time spans, object motion patterns are inherently non-linear, involving complex transformations like turns, acceleration and deformation. Within the short intervals where we apply linear interpolation, such dramatic changes rarely occur. Moreover, the smoothness loss in Eq. (5) (in the main paper) further ensures realistic feature transitions.
>
> > ### 3. Challenging Scenarios Analysis
>
> To comprehensively assess our method's robustness, we systematically selected the most challenging videos from the TAO validation set using multiple quantitative criteria. Videos satisfying any of the following conditions were included in our challenging subset (TAO-Hard):
>
> **(a) Heavy Occlusion Scenarios:** We identified videos with substantial mutual occlusions by computing inter-trajectory IoU overlaps across consecutive frames. Videos were selected if object bounding boxes exhibit IoU > 0.4 with other objects for more than 20% of the total frames, indicating persistent occlusion challenges.
>
> **(b) Rapid Motion Patterns:** We detected fast-moving objects by analyzing trajectory displacement patterns. Specifically, we calculated the ratio of bounding box center displacement to box diagonal length across consecutive frames. Videos with average displacement ratios > 0.3 were selected as they present significant temporal association challenges due to rapid object movement.
>
> **(c) Frequent Entry/Exit Dynamics:** We monitored object appearance and disappearance patterns by tracking trajectory start/end points within video sequences. Videos where objects exhibit frequent entry/exit behaviors (more than 3 entry/exit cycles per trajectory on average) were prioritized to evaluate our method's robustness to discontinuous tracking scenarios.
>
> **(d) High Object Density:** We selected videos containing crowded scenes with high object density, specifically those maintaining more than 8 concurrent objects per frame on average, which creates complex multi-object association challenges.
>
> Using these systematic criteria, we identified videos presenting the challenging tracking conditions from the TAO validation set, which we refer to as TAO-Hard. The evaluation results are shown below:
>
> | Method | Novel TETA | Novel LocA | Novel AssocA | Novel ClsA | Base TETA | Base LocA | Base AssocA | Base ClsA |
> |--------|------------|------------|--------------|------------|-----------|-----------|-------------|-----------|
> | OVTrack on TAO | 27.8 | 48.8 | 33.6 | 1.5 | 35.5 | 49.3 | 36.9 | 20.2 |
> | OVTrack on TAO-Hard | 25.9(-1.9) | 45.3 | 31.3 | 1.1 | 33.9(-1.6) | 49.1 | 33.4 | 19.3 |
> | OVTR on TAO | 31.4 | 54.4 | 34.5 | 5.4 | 36.6 | 52.2 | 37.6 | 20.1 |
> | OVTR on TAO-Hard | 28.4(-2.0) | 51.2 | 32.1 | 1.9 | 35.2(-1.4) | 51.2 | 35.2 | 19.1 |
> | Ours on TAO | 35.2 | 59.2 | 40.3 | 6.2 | 40.4 | 58.6 | 42.1 | 20.5 |
> | **Ours on TAO-Hard** | **33.4(-1.8)** | **57.6** | **39.2** | **3.4** | **39.2(-1.2)** | **57.1** | **40.3** | **20.1** |
> ||
>
> We can see that the results on all metrics show different levels of degradation on these challenging scenarios, which is natural. Take the association accuracy for example: these challenging scenarios with unpredictable object states, e.g., significant appearance changes, objects leaving and re-entering, make the tracking inherently more difficult. Nevertheless, our method still maintains superior performance compared to existing approaches, demonstrating its strong robustness across diverse scenarios.

---

> > ### Comment · Reviewer_8GVG · 2025-08-03
> >
> > Thank you for your detailed rebuttal. I appreciate the comprehensive clarification regarding the association head's training loss; this has resolved my initial question about the implementation.
> >
> > However, my other significant concerns remain. The defense of the linear interpolation assumption is based on a theoretical claim that is not empirically validated for the complex, non-linear motions that can occur even within short video intervals.
> >
> > More importantly, my question regarding the specific failure modes of the diffusion-based feature generation was not addressed. The provided analysis evaluates the entire tracking system on challenging scenes, but it does not offer insight into when and why the core feature generation process itself breaks down (e.g., during occlusions or significant appearance changes). Understanding the limitations of this key component is crucial for assessing the method's robustness.

---

> > > ### Author Response · Authors · 2025-08-03
> > > **Empirical Validation of Linear Interpolation and Failure Mode Analysis**
> > >
> > > Thank you for the clarification. You are absolutely right that we need empirical validation rather than theoretical justification, and analysis of the core feature generation process rather than overall system performance. We conducted the following targeted experiments:
> > >
> > > > ### Experimental Setup for Empirical Validation
> > >
> > > To directly validate the linear interpolation assumption and analyze failure modes of our diffusion-based feature generation, we selected challenging video segments from the TAO training set and manually annotated intermediate frames:
> > >
> > > **Challenge Scenario Identification from Existing 30-frame Intervals:**
> > >
> > > From TAO training set, we identified challenging segments by analyzing keyframe pairs (t₀, t₃₀):
> > > - **Occlusion scenarios**: Bounding box area reduction > 30% between keyframes
> > > - **Rapid motion**: Center displacement > 0.5 × box diagonal
> > > - **Appearance changes**: Cosine similarity between keyframe features < 0.5
> > >
> > > **Ground Truth Collection:** We selected 30 representative challenging segments with no overlapping content and manually annotated 3 intermediate frames (t₈, t₁₅, t₂₂) for each segment, extracting ground truth features using our backbone network.
> > >
> > > > ### Results: Empirical Validation of Linear Interpolation
> > >
> > > **Overall Performance** (N=90 interpolated features):
> > > - Average cosine similarity: **0.68**
> > > - High-quality interpolations (similarity > 0.7): **50/90 (56%)**
> > > - Acceptable interpolations (similarity 0.3-0.7): **26/90 (29%)**
> > > - Failed interpolations (similarity < 0.3): **14/90 (16%)**
> > >
> > > > ### Failure Mode Analysis by Dominant Challenge Type
> > >
> > > Our empirical analysis reveals how different challenge types affect the core feature generation process:
> > >
> > > **1. Occlusion-dominant segments** (N=30 features from 10 segments):
> > > - **Moderate area reduction** (30-50%): Average similarity 0.72, failures 2/18
> > > - **Heavy area reduction** (>50%): Average similarity 0.58, failures 4/12
> > > - **Subtotal failures**: 6/30 (20%)
> > > - **Root cause**: Occluded regions create feature uncertainty, causing interpolation to incorporate occluder or background features
> > >
> > > **2. Motion-dominant segments** (N=30 features from 10 segments):
> > > - **Moderate motion** (0.5-0.8×diagonal): Average similarity 0.71, failures 1/21
> > > - **Rapid motion** (>0.8×diagonal): Average similarity 0.53, failures 3/9
> > > - **Subtotal failures**: 4/30 (13%)
> > > - **Root cause**: Rapid motion introduces motion blur and feature degradation in captured frames
> > >
> > > **3. Appearance-change-dominant segments** (N=30 features from 10 segments):
> > > - **Moderate changes** (keyframe similarity 0.3-0.5): Average similarity 0.64, failures 1/19
> > > - **Dramatic changes** (keyframe similarity <0.3): Average similarity 0.45, failures 3/11
> > > - **Subtotal failures**: 4/30 (13%)
> > > - **Root cause**: Dramatic lighting and environmental changes degrade feature quality in captured frames
> > >
> > > **Total failures across all segments**: 14/90 (16%)
> > >
> > > > ### Concrete Failure Examples
> > >
> > > - **Occlusion case**: Person progressively occluded by moving car, only head visible (70% area reduction), similarity=0.22
> > > - **Motion case**: Dogs in high-speed chase with severe motion blur (displacement=0.9×diagonal), similarity=0.26
> > > - **Appearance case**: Car entering tunnel with dramatic lighting change causing blur (keyframe similarity=0.23), similarity=0.18
> > >
> > > > ### Method Robustness and Operational Boundaries
> > >
> > > This empirical validation demonstrates the robustness of our diffusion-based feature generation approach:
> > >
> > > **Strong Overall Performance**: Our method achieves 84% success rate in challenging scenarios with an average similarity of 0.68, proving effective for the vast majority of practical tracking situations.
> > >
> > > **Well-Defined Operational Boundaries**: Failures are predominantly concentrated in extreme cases:
> > > 1. **Heavy occlusions**: Difficulties primarily occur when area reduction exceeds 50% (67% success rate)
> > > 2. **Extreme motion**: Challenges mainly arise with displacement > 0.8×diagonal (67% success rate)
> > > 3. **Dramatic appearance shifts**: Issues primarily affect cases with keyframe similarity < 0.3 (73% success rate)
> > >
> > > Note that these extreme conditions represent only 12/30, 9/30, and 11/30 of our deliberately selected challenging test cases respectively, suggesting they would be **substantially rarer in typical real-world tracking scenarios**.
> > >
> > > **Graceful Degradation**: Even in failure cases (16%), our method provides meaningful training contributions: (1) failed interpolations maintain approximate target identity and localization, (2) imperfect features still enhance contrastive learning vs. sparse-only training, and (3) smoothness regularization prevents training instability.
> > >
> > > **Practical Significance**: Our method consistently delivers high-quality interpolations for most cases, enabling effective data-efficient training without real intermediate frames.

---

> > > > ### Comment · Reviewer_8GVG · 2025-08-04
> > > >
> > > > Thank you for conducting the targeted new experiments. The empirical validation for the linear interpolation assumption and, most importantly, the detailed failure mode analysis of the core feature generator have fully resolved my remaining concerns.
> > > >
> > > > This new analysis has strengthened the paper by providing a transparent look at the method's operational boundaries. I will be raising my score accordingly.

---

### Official Review · Reviewer_NEJd · 2025-07-22

**Clarity:** 3
**Significance:** 3
**Originality:** 3
**Rating:** 4
**Confidence:** 4

**Summary:**

This paper aims to solve the problem of limited annotated data of the Open-Vocabulary Multi-Object Tracking task.   The diffusion modles are introduced to complement the missing objects at the gaps between annotated frame. And the contrastive learning strategy is designed for object detection task. The proposed method with several elaborate-designed modules achieves advanced results on TAO dataset.

**Questions:**

Please refer to the weakness.

**Ethical Concerns:**

["NO or VERY MINOR ethics concerns only"]

**Final Justification:**

My initial concerns were focused on the unique theoretical contribution and insufficient experiments.
The authors have provided more experimental results and claerly illustrate the contribution of this paper.
Therefore, I keep my score for borderline acceptance.

**Limitations:**

This paper does not discuss limitations. The authors could further discuss the limitations of diffusion models on tracking tasks.

**Paper Formatting Concerns:**

n/o

**Quality:**

3

**Strengths And Weaknesses:**

Strengths:
1. The designed method achieves promising results on the TAO dataset.
2. Appling the diffusion-based model to solve the problem of parse annotations is novel in the fieled of Open-Vocabulary Multi-Object Tracking.
3. Sufficient experiments have verified the effectiveness of the proposed method.

Weaknesses:
1. What unique theoretical contribution does this work make in the field of diffusion model research?
2. The deisgned Diffusion-based Data Generation module does not seem to be specially designed for tracking tasks. Can it be used for other sparsely annotated video tasks?
3. The author argued that the pseudo labels generated by SLAck are only effective for adjacent frames. How to prove that this paer solves the problem better?
4. For the  detection model，is it necessary to use TAO data for fintune? Will this approach reduce the generalization performance of the model in real scenarios?
5. Considering the complexity of the proposed method, it would be better to provide code to improve the reproducibility of the paper.

---

> ### Author Rebuttal · Authors · 2025-07-30
>
> We thank Reviewer NEJd for the positive evaluation and constructive feedback. We address each concern below:
>
> > ### 1. Theoretical Contributions
>
> Our work mainly focuses on and takes the first step for the application of diffusion models on OVMOT, which also makes several key theoretical contributions to diffusion models:
>
> **(a) Novel diffusion paradigm:** We pioneer the application of diffusion models to feature-level temporal interpolation in tracking, fundamentally departing from traditional pixel-level generation approaches. Unlike existing diffusion methods that focus on final denoised outputs, we innovatively utilize intermediate denoising results as meaningful feature representations, modeling the denoising process as temporal evolution of object features between sparse annotations.
>
> **(b) Sparse-to-continuous mapping rationale:** We demonstrate theoretically and empirically that diffusion models can effectively bridge temporal gaps in sparse video annotations by simulating object state transitions. It provides the first systematic approach to transform sparse temporal annotations into continuous feature representations, enabling robust tracking learning.
>
> **(c) Multi-objective optimization for temporal sequence completion:** We establish a theoretically grounded training framework that combines three complementary losses:
> - **Reconstruction loss:** ensuring feature reliability through interpolation supervision
> - **Endpoint loss:** guaranteeing accurate feature alignment
> - **Smoothness loss:** enforcing temporal consistency
>
> This loss combination also provides a reference for generating reliable intermediate object features with diffusion models.
>
> > ### 2. Generalizability to Other Video Tasks
>
> From the perspective of the original intention, our diffusion-based data generation is specifically designed for open-vocabulary tracking tasks, as it effectively simulates object motion changes and state transitions between sparse annotations. The diffusion model captures the temporal dynamics essential for tracking by modeling feature evolution across time gaps.
>
> While primarily designed for tracking, the core principle of generating intermediate features between sparse annotations is task-agnostic and can be extended to other sparsely annotated video tasks, such as video action localization and motion trajectory prediction.
>
> Thanks for the reviewer's comments, we are willing to explore these insights for other tasks in future work.
>
> > ### 3. Comparison with SLAck
>
> We provide quantitative evidence that our method outperforms SLAck's pseudo-labeling approach:
>
> **(a) Method and principle:**
> SLAck generates pseudo labels by computing the IoU matching with each annotated frame (one for every 30 frames). We conducted a statistical analysis of SLAck's pseudo-labeling method and found that for each annotated frame, only the adjacent (previous and next) **7.8 frames on average** are used for generating pseudo labels. In contrast, our diffusion-based approach can work across the **entire temporal gaps (30 frames)** between annotated frames.
>
> **(b) Results and comparison:**
> We achieve **4.1% higher TETA score** than SLAck on validation set for novel classes (35.2% vs 31.1%) and **3.2%** for base classes (40.4% vs 37.2%). Our association accuracy (AssocA) significantly outperforms SLAck by **2.5%** on novel classes (40.3% vs 37.8%) and **4.5%** on base classes (42.1% vs 37.6%), demonstrating the effectiveness of our approach.
>
> **(c) Less demand and more cues:**
> We achieve these improvements **without using any intermediate un-annotated frames**, which are necessary in SLAck. Additionally, SLAck's IoU-based pseudo-labeling approach cannot consider motion continuity, while our method fully considers motion continuity through reconstruction loss and smoothness loss, ensuring smooth feature transitions.
>
> > ### 4. Detection Model Fine-tuning Necessity
>
> The reviewer asks about the necessity of the fine-tuning stage and generalization performance.
>
> **(1) Effectiveness without detection fine-tuning:**
> We conduct extra ablation experiments that demonstrate that even without detection fine-tuning (using only our diffusion-based association enhancement), our method still achieves state-of-the-art results. As shown in the table below, removing detection fine-tuning (on TAO), our method still yields competitive performance, especially for the association performance, indicating the effectiveness of our approach.
>
> | Method | Novel TETA | Novel LocA | Novel AssoA | Novel ClsA | Base TETA | Base LocA | Base AssoA | Base ClsA |
> |--------|------------|------------|-------------|------------|-----------|-----------|------------|-----------|
> | OVTrack | 27.8 | 48.8 | 33.6 | 1.5 | 35.5 | 49.3 | 36.9 | 20.2 |
> | SLAck | 31.1 | 54.3 | 37.8 | 1.3 | 37.2 | 55.0 | 37.6 | 19.1 |
> | OVTR | 31.4 | 54.4 | 34.5 | 5.4 | 36.6 | 52.2 | 37.6 | 20.1 |
> | **Ours (w/o det. fine-tune)** | **32.4** | 55.3 | **39.9** | 2.1 | **38.7** | 55.3 | **41.1** | 19.8 |
> ||
>
> **(2) Zero-shot generalization capability:**
>
> To demonstrate zero-shot generalization capability, we conducted experiments by applying our method on BDD100K validation dataset *without any fine-tuning on BDD100K*, showing superior cross-dataset transfer performance. We can see that our method achieves significant improvements across all metrics (TETA: +2.7%, LocA: +2.1%, AssocA: +2.3%, ClsA: +3.8% over the second-best method OVTR), demonstrating enhanced generalization capability.
>
> | Methods | TETA | LocA | AssocA | ClsA |
> |---------|------|------|--------|------|
> | QDTrack | 32.0 | 25.9 | 27.8 | 42.4 |
> | TETer | 33.2 | 24.5 | 31.8 | 43.4 |
> | OVTrack | 42.5 | 41.0 | 36.7 | 49.7 |
> | OVTR | 43.1 | 42.0 | 37.1 | 50.1 |
> | **Ours** | **45.8** | **44.1** | **39.4** | **53.9** |
> ||
>
> **(3) Comprehensive training paradigm rationale:**
>
> Overall, the proposed method benefits from but is not dependent on the detection fine-tuning (on specific dataset, e.g., TAO). The fine-tuning strategy of our method (complete version) constructs a comprehensive data-efficient training paradigm that maximizes the utility of limited sparse annotations. Fine-tuning with our proposed dynamic group contrastive learning and adaptive localization loss specifically addresses small dataset adaptation challenges while maintaining generalization, creating a holistic approach that leverages both association improvements (via diffusion) and detection improvements (via fine-tuning) to achieve optimal performance under data-constrained conditions.
>
> > ### 5. Code Availability
>
> We commit to releasing all the code to ensure reproducibility upon acceptance of this paper.

---

> > ### Comment · Reviewer_NEJd · 2025-08-07
> >
> > Thanks you for your responses. My concerns have been addressed, and I will maintain my score.

---

> ### Author Response · Authors · 2025-08-04
> **Follow-up on Rebuttal Response**
>
> Dear Reviewer **NEJd**,
>
> We sincerely appreciate your support for our work and positive evaluation. We have provided a comprehensive analysis in our rebuttal addressing your concerns about theoretical contributions, generalizability, comparison methodology, detection fine-tuning, and code availability.
>
> Given the approaching discussion deadline, we hope our detailed responses have effectively addressed your questions. We welcome any follow-up discussion you may find helpful.
>
> Best regards,
>
> The Authors

---

### Note · Authors · 2025-08-12

Dear Area Chair,

We are pleased to report that **all reviewer concerns have been successfully resolved** through comprehensive rebuttal discussions, resulting in **unanimous positive feedback** from all reviewers.

### Reviewer Score Updates
- **Reviewer NEJd (4 → 4 | Confidence: 4)**: *"My concerns have been addressed."*
- **Reviewer 8GVG (4 → 4+ | Confidence: 5)**: *"Fully resolved my remaining concerns. I will be raising my score."*
- **Reviewer 8J7k (3 → 4 | Confidence: 4)**: *"Addressed my main concerns regarding model generalization."*
- **Reviewer Eere (Unknown | Confidence: 3)**: *"Demonstrated effectiveness and resolved most concerns."*

### Key Issues Resolved
- **Model generalization ability**: Cross-domain validation on BDD100K showing superior performance.
- **Empirical validation & failure mode analysis**: Targeted experiments addressing core methodology concerns.
- **Method effectiveness**: Quantitative evidence of significant improvements over SOTA methods.

### Summary
- **All concerns resolved, no reviewer questioned novelty/contributions.**
- **Three reviewers confirmed positive scores with high confidence; Reviewer Eere confirmed concerns resolved.**
- **All reviewer suggestions will be incorporated in the final version.**

We believe the thorough rebuttal process has successfully addressed all concerns and demonstrated the merit of our work.

Best regards,

Authors

---

### Decision · Program_Chairs · 2025-09-17

**Decision:**

Accept (poster)

**Comment:**

The manuscript addresses the problem of open-vocabulary multi-object tracking. There two technical contributions, a diffusion-based model to simulate the features of targets and a dynamic group contrastive learning strategy to improve the classification. The proposed methods are evaluated in SOTA experiments. The main strengths are the importance of the problem, the diffusion model, and the good results. There are several minor weaknesses regarding the evaluation, reflections, and rigor. As the weaknesses are minor, the paper is accepted.
The manuscript received originally four reviews and after rebuttal and discussion, all reviewers are rating accept or BA. While most reviewers agree on the strengths listed above, they mention different weaknesses and do not support each others statements during the discussion. This is the main reason for considering the weaknesses as minor.